



# The synergistic impact of ENSO and IOD on the Indian Summer Monsoon Rainfall in observations and climate simulations - an information theory perspective

Praveen Kumar Pothapakula[1], Cristina Primo[1], Silje Sørland [2], and Bodo Ahrens [1]

[1]Institute for Atmospheric and Environmental Sciences, Goethe University Frankfurt am Main, Germany.
[2]Dep. of Environmental Systems Science, ETH Zürich, Switzerland.

**Correspondence:** Praveen Kumar Pothapakula (pothapakula@iau.uni-frankfurt.de)

**Abstract.** El-Niño southern oscillation (ENSO) and Indian Ocean Dipole (IOD) are two well-know temporal oscillations in the sea surface temperature (SST), which both are thought to influence the interannual variability of the Indian Summer Monsoon Rainfall (ISMR). Until now, there has been no measure to assess the simultaneous information exchange (IE) from both ENSO and IOD to ISMR. This study explores the information exchange from two source variables (ENSO and IOD) to

one target (ISMR). First, in order to illustrate the concepts and quantification of two-source IE to a target, we use idealized test cases consisting of linear as well as non-linear dynamical systems. Our results show that these systems exhibit net synergy (i.e., the combined influence of two sources on a target is greater than the sum of their individual contributions), even with uncorrelated sources in both the linear and non-linear systems. We test IE quantification with various estimators (the Linear, Kernel, and Kraskov estimators) for robustness. Next, the two-source IE from ENSO and IOD to the ISMR is investigated in

observations, reanalysis, three global climate model (GCM) simulations, and three nested, higher-resolution simulations using a regional climate model (RCM). This (1) quantifies IE from ENSO and IOD to ISMR in the natural system, and (2) applies IE in the evaluation of the GCM and RCM simulations. The results show that both ENSO and IOD contribute to the ISMR interannual variability. Interestingly, significant net synergy is noted in the central parts of the Indian subcontinent, which is India's monsoon core region. This indicates that both ENSO and IOD are synergistic predictors in the monsoon core region.

But, they share significant net redundant information in the southern part of Indian subcontinent. The IE patterns in the GCM simulations differ substantially from the patterns derived from observations and reanalyses. Only one nested RCM simulation IE pattern adds value to the corresponding GCM simulation pattern. Only in this case, the GCM simulation shows realistic SST patterns and moisture transport during the various ENSO and IOD phases. This confirms, once again, the importance of the choice of the GCM in driving a higher-resolution RCM. This study shows that two-source IE is a useful metric that helps

in better understanding the climate system and in process-oriented climate model evaluation.

## 1 Introduction

The South Asian Monsoon is considered as a large-scale coupled air-sea-land interaction phenomenon that brings seasonal rainfall to the Indian subcontinent and other near areas (Webster et al, 1988). Large parts of the Indian subcontinent receive



rainfall from June to September known as the Indian Summer Monsoon Rainfall (ISMR). The ISMR contributes about 70–90%

to the total annual precipitation amount in the Indian subcontinent (Shukla and Haung, 2016). The agriculture in the Indian subcontinent depends substantially on the ISMR, and any variations on the interannual as well as intraseasonal variabilities of ISMR cause a significant impact on the country's economy. The interannual variation of the IMSR is only about 10% of the mean (Gadgil, 2003), yet it has a large impact on crop production. The mean seasonal rainfall predictability significantly depends on the interannual variability of the ISMR (Goswami et al., 2006a; Pillai and Chowdary, 2016). The interannual

variability of the ISMR is linked to many noted oscillations, the El Niño Southern Oscillation (ENSO), Indian Ocean Dipole (IOD), Atlantic Multidecadal Oscillation (AMO), Atlantic Zonal Mode (AZM), Pacific Decadal Oscillation (PDO), etc., (Nair et al., 2018; Sabeerali et al., 2019; Hrudaya et al., 2020). The oscillations thought to have the most significant impact on the ISMR are ENSO and IOD (Krishnaswami et al., 2015). Hence, in this study, we majorly focus on the individual and combined influences of the two climate modes ENSO and IOD on the ISMR interannual variability in observations, reanalysis data sets,

and climate models.

ENSO is an important large-scale coupled atmosphere-ocean aperiodic oscillation over the Pacific ocean that on average occurs every 2–7 years. The Sea Surface Temperature (SST) pattern over western (central-eastern) tropical Pacific ocean experience large cold (warm) anomalies during the El Niño phase. The normal patterns of SST over the Pacific ocean are enhanced during the La Niña phase. These variabilities in the SST are coupled to the atmospheric Walker circulation, and Sir

Gilbert Walker in 1924 was the first to observe a relation between ENSO and ISMR (Walker, 1924; Gadgil, 2003; Goswami, 1998; Yun and Timmermann, 2018). He noticed that often the El Niño (La Niña) conditions over the Pacific ocean are linked to weak (strong) ISMR. During the El Niño conditions, the entire walker circulation is shifted eastwards by which the descending branch of the Walker cell on the western Indian ocean shifts eastward to overlie on the Indian subcontinent, thereby suppressing the convection (Walker, 1924; Krishna Kumar et al., 2006; Palmer et al., 2006). In the La Niña years, the entire Walker

circulation shifts slightly westward, which assists in enhancing the convection over the Indian subcontinent. Many other studies (Goswami, 1998; Slingo and Annamalai, 2000) argued that the El Niño conditions do not suppress the ISMR directly through the descending branch of the Walker circulation but rather, the changes in the Walker circulation enhances the meridional Hadley circulation decent over the Indian subcontinent. Hence it could be that the ENSO affects the IMSR through interactions between the Walker and Hadley circulations.

Another important source that is linked to the ISMR interannual variability is a dipole like structure in the Indian ocean surface temperature known as IOD (Saji et al., 1999). During a positive (negative) IOD, the southeastern part of the Indian ocean is cooler (warmer) than normal while the western part of the Indian ocean is warmer (cooler). During the positive IOD event, the meridional circulation in the region is modulated through anomalous convergence patterns over the Bay of Bengal, thereby strengthening the monsoon with anomalous positive rainfall over the Indian subcontinent while the negative IOD events

lead to the weakening of the rainfall (Ashok et al., 2001). Behera and Ratnam (2018) found that the opposite phases of IOD are associated with distinct regional asymmetries in IMSR anomalies over the Indian subcontinent contributing significantly to the interannual variability. Interestingly, Ashok et al. (2001) found that during the co-existence of El Niño and positive IOD, the IOD tends to compensate for the influence of El Niño leading to normal rainfall by inducing anomalous convergence over the



Bay of Bengal. Similarly, the negative IOD events can reduce the impact of La Niña on ISM rainfall and cause deficit monsoon

rainfall. However, the study of Chowdary et al. (2015) showed that the local air–sea interaction in the tropical Indian ocean

opposes the Pacific ocean impact even in the absence of IOD. Hence, still there are uncertainties associated with the individual

and combined influence of ENSO and IOD on the interannual variability of ISMR.

Motivated by these large uncertainties in the present knowledge about how ENSO and IOD influence the ISMR interannual

variability, we are investigating these connections from a two-source information exchange (IE) perspective. Shannon (1948)

first introduced the concept of information entropy, which quantifies the average uncertainty of a given random variable. The IE

between two subsystems $X$ and $Y$ can be understood as the average uncertainty reduction about $X$ in knowing $Y$ or vice versa.

Recently, various methods from information theory have been widely used in the fields of earth system sciences (Bennett et al.,

2019; Gerken et al., 2019; Jiang and Praveen, 2019; Ruddel et al., 2019), climate sciences (Nowack et al., 2020; Runge et al.,

2019; Joshua et al., 2019; Campuzano et al., 2018; Bhaskar et al., 2017) and in other interdisciplinary sciences (Wibral et al.,

2017; Leonardo et al., 2019; Shoaib Ahmad, 2018). The information theory, in its current form, provides a complete description

of the IE relationship between a single-source and a target. However complex climate system often consist of multi-sources

influencing a target such as the ENSO and IOD influencing the ISMR variability.

The IE in a system composed of two-source systems $Y$ and $Z$ to the target variable $X$ is decomposed into four parts (Fig.

1) according to Williams and Beer (2010): (i) unique information shared by $Y$ to $X$ (ii) unique information shared by $Z$ to

$X$ (iii) redundant information or overlapping information shared by both sources $Y$ and $Z$ together with $X$ (iv) synergistic

information about $X$ while knowing $Y$ and $Z$ together but not either of them alone. An example of synergistic information

from two sources is the classical binary exclusive-or (XOR) operation (Williams and Beer, 2010; James et al., 2016), where

the two sources $Y$ and $Z$ provide information that is not available from either of their states alone but by jointly knowing

their states together. Since ENSO and IOD are known to simultaneously influence the ISMR variability, one could expect

the component of synergy or redundant information existing in this climate phenomenon. In the case of synergy, the target

uncertainty of IMSR interannual variability is reduced only when the states of two sources, ENSO and IOD are known together

but not individually. This decomposition of information is known as partial information decomposition (PID). Unfortunately,

with the present standard methods available from information theory, one can not obtain the contributions of unique, synergy,

and redundant information exchange metrics solely (Barrett, 2015). Here, we would like to bring to the attention of the readers

that many interesting studies have come up with various definitions of these metrics (Williams and Beer, 2010; Griffith and

Koch, 2014; Bertschinger et al., 2014; Finn and Lizer, 2018) and still, there has been no consensus among the scientific

community for obtaining these metrics. A complete and consistent framework on quantifying the individual contributions of

various terms in PID would make information theory a complete framework for understanding the information dynamics of

multi-source systems. However, with the present available information theory methods, one can obtain a metric known as net

synergy, which is synergistic information minus redundant information carried by the two sources $Y$ and $Z$ about the source $X$.

More details of the formula of net synergy are described in the data and methodology section. It is very important to note that,

though the methods from information theory are very useful in analyzing the complex system behavior, their estimations are

quite challenging due to their sensitivity to free tuning parameters and sample size (Knuth et al., 2013; Smirnov Dmitry, 2013;





Pothapakula et al., 2019). Hence, this study follows and uses various estimators we proposed in our earlier work (Pothapakula
et al., 2019) for robustness in the results.

Here we are investigating the information exchange from ENSO and IOD to the IMSR interannual variability by using available observations, reanalysis data sets, and climate models. However, before exploring the two-source IE from the ENSO and IOD to IMSR variability, we first demonstrate the concept of two-source IE with results from a simple idealized linear and non-linear dynamical models for better understanding. We also use various estimators of IE, for example, Linear, Kraskov, and
Kernel estimators for robustness. Then, the two-source IE concept is applied to observations and reanalysis data sets. This helps in understanding the IE dynamics of ENSO and IOD to the interannual variability of IMSR in the natural system. Thereafter, we investigate if the two-source information exchange dynamics of ENSO and IOD to ISMR interannual variability is replicated in three different global climate models (GCM) simulations from the 5th phase of the Coupled Model Intercomparison Project (CMIP5). Since it is well known that GCMs due to their low spatial resolution do not resolve all the subgrid-scale phenomena,
we have used dynamical downscaling of the three GCM simulations with an RCM to obtain higher resolution details (Bhaskaran et al., 2012; Chowdary et al., 2018; Dobler and Ahrens, 2011; Asharaf and Ahrens, 2015; Lucas-Picher et al., 2011). The RCM simulations are performed with a horizontal resolution of 25km ($\sim$ 0.22) and follow the framework of coordinated regional downscaling experiments (CORDEX) (Giorgi et al., 2009; Gutowski et al., 2016). By employing the two-source IE from the ENSO and IOD to the ISMR interannual variability on both the driving GCM simulations and the downscaled RCM
simulations, we can evaluate the performance of the model chain. To our knowledge, this is a first of its kind evaluation study of GCM simulations and RCM simulations with information theory methods from the two-source IE viewpoint.

This paper is organized as follows. In Section 2 we explain briefly the information theory methods and estimators used in this study followed by a brief discussion about the idealized linear and non-linear dynamical systems. In Section 3 observational and reanalysis data, various GCMs in CMIP5 used in this study, and the RCM model used in dynamically downscaling the
GCM simulations are discussed. In Section 4, the results obtained from idealized systems and model evaluation are shown along with a detailed discussion. Finally, conclusions are drawn in Section 5.

## 2   The theory of information exchange

This section comprises of the basic concepts of information theory along with a brief introduction of various estimators. Also, a description of the idealized systems used in this study is covered.

### 120   2.1   Concepts from Information Theory

The Shannon entropy (Shannon, 1948) of a random variable $X$, quantifies the amount of uncertainty contained in it and is defined by

$$H(X) = -\sum_x p(x) \log p(x),$$

where $p(x)$ is the probability of a discrete state of the random variable $X$. The summation goes through all states of the random
variable $X$. The units of entropy are expressed in nats if a natural logarithm is applied (in bits when the logarithm base is 2).





Mutual information (MI) quantifies the reduction in the uncertainty of one random variable given knowledge of another variable (Cover and Thomas, 1991) and is defined by

$$I(X;Y) = \sum_{x,y} p(x,y) \log \frac{p(x,y)}{p(x)p(y)},$$

where $p(x,y)$ is the joint distribution of variables $X$ and $Y$, and $p(x)$, $p(y)$ are the marginal distributions of $X$ and $Y$, respectively.

Mutual information between two sources $Y$ and $Z$ and a target $X$ is given as

$$I(X;Y,Z) = \sum_{x,y,z} p(x,y,z) \log \frac{p(x,y,z)}{p(x)p(y,z)},$$

where $p(x,y,z)$ is the joint distribution of variables $X,Y$ and $Z$, and $p(x)$, $p(y,z)$ are the marginal probabilities. Furthermore, the information $I(X;Y,Z)$ that the two sources share with target should decompose according to partial information decomposition by Williams and Beer (2010) into four parts (Fig. 1) as

$$I(X;Y,Z) = U(X;Y|Z) + U(X;Z|Y) + R(X;Y,Z) + S(X;Y,Z), \tag{1}$$

where $U(X;Y|Z)$ is the unique information shared by $Y$ to $X$, $U(X;Z|Y)$ is the unique information shared by $Z$ to $X$, $R(X;Y,Z)$ redundant information shared by both sources $Y$ and $Z$ together with $X$, and $S(X;Y,Z)$ synergistic information about $X$ while knowing the states of $Y$ and $Z$ together.

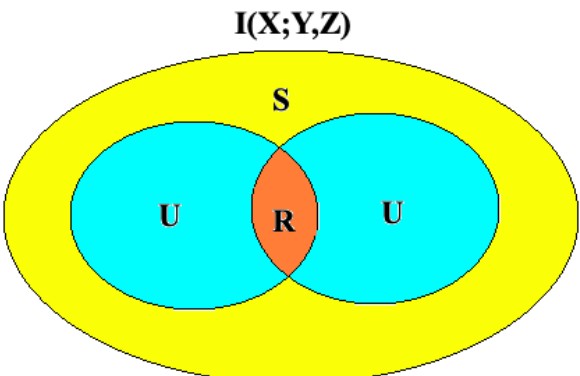

**Figure 1.** Information exchange from two sources $Y$, $Z$ to the target $X$ decomposed according to PID as unique information (U), redundant information (R) and synergistic information (S)





In the case of two sources influencing the target, the mutual information shared by a single source to the target is given by

$$I(X;Y) = U(X;Y|Z) + R(X;Y,Z),$$
$$I(X;Z) = U(X;Z|Y) + R(X;Y,Z).$$
(2)

From the current information theory framework, the quantities $I(X;Y,Z)$, $I(X;Y)$, $I(X;Z)$ can be straightforwardly computed. However, there are still ongoing debates about quantifying unique information, redundant information, and synergistic information.

According to Barrett (2015), one can obtain a quantity known as net synergy from Eq.1 and Eq.2 as

$$\Delta I(X;Y,Z) = I(X;Y,Z) - I(X;Y) - I(X;Z),$$
$$= S(X;Y,Z) - R(X;Y,Z).$$
(3)

When $\Delta I(X;Y,Z) > 0$, synergistic information from two sources is greater than redundant information and vice versa. The $\Delta I$ provides an lowerbound for synergistic/redundant information. From here on, if $\Delta I(X;Y,Z) > 0$ we refer as net synergistic information and if $\Delta I(X;Y,Z) < 0$ we refer to as net redundant information.

## 2.2  Estimation techniques

Though the information theory methods are very useful in assessing the behavior of dynamical systems, their estimation is

challenging. Hence, in this study, we implemented various estimators for robustness in our results.

### 2.2.1  Estimation under linear approximation (Linear estimator)

Here we will briefly introduce the basic concepts for estimation of the two-source IE under linear approximation. For a detailed explanation of the concept, we are referring the reader to Barrett (2015).

The entropy for a continuous random variable $X$ under linear approximation is given as


$$H(X) = \frac{1}{2}\log[\det \Sigma(X)] + \frac{1}{2}m\log(2\pi e),$$

where $m$ is the dimension of random variable $X$, $\Sigma(X)$ is the $m \times m$ matrix covariances i.e., $\mathrm{cov}(X^i, X^j)$.

Following Barrett (2015), the partial covariance of $X$ with respect to $Y$ is given as

$$\Sigma(X|Y) = \Sigma(X) - \Sigma(X,Y)\Sigma(Y)^{-1}\Sigma(Y,X).$$

From then the conditional entropy can be derived as

$$H(X|Y) = \frac{1}{2}\log[\det \Sigma(X|Y)] + \frac{1}{2}m\log(2\pi e).$$

The mutual information $I(X;Y)$ is the difference between $H(X)$ and $H(X|Y)$,

$$I(X;Y) = \frac{1}{2}\log\left[\frac{\det \Sigma(X)}{\det \Sigma(X|Y)}\right].$$





For a general three dimensional jointly Gaussian system $(X,Y,Z)^T$, and by setting zero mean and unit variance, the covariance matrix is given by,

$$\Sigma = \begin{bmatrix} 1 & a & c \\ a & 1 & b \\ c & b & 1 \end{bmatrix}$$
.

Thus, from the above matrix, the mutual information is given as

$$I(X;Y) = \frac{1}{2}\log\left(\frac{1}{1-a^2}\right),$$

$$I(X;Z) = \frac{1}{2}\log\left(\frac{1}{1-c^2}\right),$$

$$I(X;Y,Z) = \frac{1}{2}\log\left(\frac{1-b^2}{1-(a^2+b^2+c^2)+2abc}\right).$$

The net synergy can be obtained by $I(X;Y,Z) - I(X;Y) - I(X;Z)$, given as

$$\Delta I(X;Y,Z) = \frac{1}{2}\log\left(\frac{(1-a^2)(1-b^2)(1-c^2)}{1-(a^2+b^2+c^2)+2abc}\right).$$

### 2.2.2 Estimation through box step kernel (Kernel estimator)

The estimation of non-linear entropy and mutual information estimators contains Probability Density Functions (PDFs). The uni-variate and bi-variate PDFs for continuous data can be estimated through various available discretization methods (e.g.,

binning, kernel etc). Here we use a simple box step kernel $\Theta$ with $\Theta(x>0)=0$ and $\Theta(x<0)=1$ for the estimation of relevant joint probability distributions (e.g., $\hat{p}(x,y)$, $\hat{p}(x)$ and $\hat{p}(y)$). For example, the joint probability distribution $\hat{p}(x,y)$ is calculated as

$$\hat{p}_r(x_n,y_n) = \frac{1}{N}\sum_{n'=1}^{N}\Theta(|(x_n-x_{n'}),(y_n-y_{n'})|-r),$$

where the norm corresponds to the maximum distance in the joint space and $r$ is the kernel width. Similarly one can estimate

the PDF for high dimensional systems for the estimation of MI. For more details into the estimator, refer to Kantz and Schreiber (1997); Goodwell and Kumar (2017) and information-theoretic toolkit from Lizier (2014).

### 2.2.3 Estimation through k-nearest neighbor (Kraskov estimator)

The k-nearest neighbor estimator uses an adaptive binning strategy by estimating the average distances to the k-nearest neighbor data points. For example, the MI can be computed as

$$I(X;Y) = \Psi(k) - < \Psi(n_x+1) + \Psi(n_y+1) > + \Psi(N),$$





where $N$ is total number of points, $n_x$ and $n_y$ are the number of points that fall in the marginal spaces of $X$ and $Y$ respectively within the distance taken as $d = \max(||x - x'||, |y - y'||)$ and $\Psi$ denotes the digamma function. For more details refer to Kraskov et al. (2004). Similarly, the equation mentioned above can be extended to higher dimensional estimation of MI. From hereafter, the estimation through k-nearest neighbor is called as Kraskov estimator.

## 2.3 Idealized systems for demonstration

Before we apply information theory estimators to two-source information exchange in climate applications, we consider idealized linear and non-linear systems as given in the following sub-sections to demonstrate the concept of two-source IE.

### 2.3.1 Linear autoregressive systems

Often in climate systems, the future state prediction of a variable relies on the past of its own state (persistence) or from past of another variable (Runge et al., 2014), or from the linear/non-linear combination of both (possible case of net synergy/redundancy). Hence, as a first case of demonstration, we considered a two-dimensional linear system (Barrett, 2015) $x$ and $y$, with $x$ receiving information from its immediate past and from the immediate past of $y$ with the following governing equations:

$$
\begin{aligned}
x_t &= \alpha x_{t-1} + \alpha y_{t-1} + \mathcal{N}_x(0,1), \\
y_t &= \mathcal{N}_y(0,1),
\end{aligned}
\tag{4}
$$

where $\alpha$ is the coupling coefficient varied from 0 to 0.8 with an increment of 0.1 and $\mathcal{N}(0,1)$ is Gaussian noise with zero mean and unit variance. The system was initialized with $(x_0 = 0)$ and is integrated around 100,000 iterations. For the analysis of two-source IE with various estimators, we use the last 5000 time units from the available time series.

In the first example, we considered IE from two sources (one source being the persistence) contributing to the target prediction, however not all predictions of target depend on two sources simultaneously (i.e., net synergy/redundancy do not exist), hence as a second case , we considered a system consisting of two subsystems which are coupled with each other but only having a single source with the governing equations

$$
\begin{aligned}
x_t &= \alpha y_{t-1} + \mathcal{N}_x(0,1), \\
y_t &= \alpha x_{t-1} + \mathcal{N}_y(0,1),
\end{aligned}
\tag{5}
$$

with $\alpha$ being the coupling coefficient. We followed similar steps for integration as in the previous linear system.

Finally as third example, we test a three-dimensional system in which two individual sub-systems contribute to the evolution of third system such as the ENSO and IOD, as two individual systems contributing to the interannual variability of the ISMR. This system has the governing equations

$$
\begin{aligned}
x_t &= \alpha y_{t-1} + \alpha z_{t-1} + \mathcal{N}_x(0,1), \\
y_t &= \mathcal{N}_y(0,1), \\
z_t &= \mathcal{N}_z(0,1),
\end{aligned}
\tag{6}
$$





where system $y$ and $z$ are two individual sub-systems exchanging information to the target system $x$.

### 2.3.2 Non-linear Heńon system

205 As the climate system is non-linear, we further extend our analysis from idealized linear systems to a idealized non-linear system. For this purpose, we considered coupled Heńon maps which captures the stretching and folding dynamics of chaotic systems such as the Lorenz system which mimic's the atmospheric behavior. We considered two Heńon maps (Kraskovska, 2019), $x$ and $y$ coupled with each other with the governing equations

$$
\begin{aligned}
x_t &= 1.4 - x_{t-1}^2 + 0.3x_{t-2}, \\
y_t &= 1.4 - (Cx_{t-1}y_{t-1} + (1-C)y_{t-1}^2) + 0.3y_{t-2},
\end{aligned}
\tag{7}
$$

210 where the coupling coefficients $C \in [0, 0.6]$. From Eq.7 it is evident that the system $x$ is driving system $y$ through coupling coefficient $C$.

## 3 Data and climate models

In this section, we will discuss various observational and reanalysis data sets used to quantify the two-source IE from ENSO and IOD to IMSR interannual variability in the natural system. Furthermore, the details of various GCM and RCM simulations 215 used in this study are also covered.

### 3.1 Observational, reanalysis data sets and climate simulations

We are focusing on the South Asian Summer Monsoon seasons, starting from June and ending in September (June- July-August-September: JJAS), thus monthly data sets for JJAS for the time period 1951-2005 from observations and model simulations are used in this study. Various observational, reanalysis data sets and model simulations used to quantify the two-source 220 IE from the ENSO and IOD to the ISMR interannual variability are listed in Table 1 and are also described here.

### 3.1.1 Observational, reanalysis data sets and indices

The UK Met Office's Hadley Centre Sea Ice and Sea Surface Temperature dataset (HadISST 1.1) (Rayner et al., 2002) is used to retrieve SST information for the Indian and the Pacific ocean. Monthly precipitation fields from Global Precipitation Climatology Centre (GPCC) (Schneider et al., 2008) is used as precipitation observational record together with a high-resolution data 225 set, covering only the monsoon south Asia domain, namely the Asian Precipitation - Highly-Resolved Observational Data Integration Towards Evaluation (APHRODITE) monthly accumulated precipitation (Akiyo et al., 2012). The rainfall, winds, and specific humidity are taken from the National Center for Environmental Prediction–National Center for Atmospheric Research (NCEP–NCAR) reanalysis data set (Kalnay et al., 1996). The ENSO and IOD indices are obtained from the National Oceanic and Atmospheric Administration Earth System Research Laboratories(NOAA ESRL) and Japan Agency for Marine-Earth Science 230 ence and Technology(JAMSTEC) for validation of PCs derived from the observational SST data sets, i.e., the HadISST, and





NCEP reanalysis SST. In addition to the above-mentioned data sets, we also used ERA-Interim (Dee et al., 2011) and MERRA (Rienecker et al., 2011) reanalysis rainfall datasets (1980-2005) as additional resources.

### 3.1.2 Global and regional climate simulations

The three CMIP5 GCMs (details in Table. 1), the MPI-ESM-LR (Stevens et al., 2017), Nor-ESM-M (Bentsen et al., 2012) and
EC-EARTH (Hazeleger et al., 2010) were dynamical downscaled with the non-hydrostatic regional climate model COSMO-crCLM version v1-1. The COSMO-crCLIM is an accelerated version of the COSMO model (Fuhrer et al., 2014) in climate mode (Leutwyler et al., 2016; Rockel et al., 2008). A two-stream radiative transfer calculations are based on Ritter and Geleyn (1992), the convection is parameterized by Tiedtke (1989), the turbulent surface energy transfer and planetary boundary layer are using the parametrization of Raschendorfer (2001), and precipitation is based on a four-category microphysics scheme that
includes cloud, rainwater, snow, and ice (Doms et al., 2011). The soil-vegetation-atmosphere-transfer is using the TERRA-ML (Schrodin and Heise, 2002), however, this current version is employing a modified groundwater formulation (Schlemmer et al., 2018). The RCM simulation has a horizontal resolution of $0.22°$ (i.e., 25km) and with 57 vertical levels and is using a time step of 150s. The model simulation configuration is following the CORDEX framework, meaning that a historical period is simulated from 1950-2005, and the business as usual future emission scenario (RCP8.5) is simulated from 2006-
2099. However, here we are only looking into the historical period. It is to be noted that for the analysis of rainfall anomaly composites, moisture anomalies, and IE plots, the GCM and RCM simulations are interpolated to a common observational grid (a grid with $0.25°$). Our interpretation of results does not change much with the original resolution of the datasets.

## 4 Results and discussion

In the current section, first, we discuss the results of two-source IE obtained from various idealized linear and non-linear
dynamical systems mentioned in Section 2. Thereafter, we present results of two-source IE in the climate system with the observations, reanalysis data sets, GCM simulations, and the RCM simulations.

### 4.1 Applications to idealized systems

First, we will start with the discussion of results obtained from idealized systems with various IE estimators.

### 4.1.1 Linear autoregressive system

Figure 2 shows the information exchange (in nats) from $y_{t-1}$ (immediate past of $y$) to $x_t$ (present of $x$) and also from $x$ immediate past to present of $x$ (i.e., $x_{t-1}$ to $x_t$), for the system with Equation 4. The two-source mutual information linear estimator shows that as the coupling coefficient increases, the IE from $I(x_t; y_{t-1}, x_{t-1})$ increases, indicating that the immediate pasts of $x_{t-1}$ and $y_{t-1}$ exchange information to the future state of $x$ as expected from the system dynamics. Also, as expected the $I(x_t; y_{t-1}, x_{t-1}) > I(x_t; y_{t-1})$ or $I(x_t; x_{t-1})$, indicating that the two-source IE dominates the dynamics of this system. The
IE from the immediate past of $x$ i.e., $x_{t-1}$ is a stronger source of information to the target $x_t$ due to self feedback/large persis-





**Table 1.** CMIP5–GCMs/RCM/observations descriptions used in the current study.

| GCM Modeling center | Acronym | Ensemble member | Atm.Resolution |
|---|---|---|---|
| Max Planck Institute for Meteorology | MPI-ESM-LR | r1i1p1 | $1.875^{\circ} \times 1.875^{\circ}$ |
| Norwegian Climate Centre | Nor-ESM-M | r1i1p1 | $2.5^{\circ} \times 1.9^{\circ}$ |
| SMHI, Sweden | EC-EARTH | r12i1p1 | $1.125^{\circ} \times 1.125^{\circ}$ |
| **RCM Modeling center** | | | |
| CLMCom-ETH | COSMO-crCLIM | | $0.22^{\circ} \times 0.22^{\circ}$ |
| **Observations and Reanalysis data sets** | | | |
| APHRODITE | – | – | $0.25^{\circ} \times 0.25^{\circ}$ |
| GPCC | – | – | $0.5^{\circ} \times 0.5^{\circ}$ |
| HadISST | – | – | $1^{\circ} \times 1^{\circ}$ |
| NCEP Reanalysis | – | – | $1.875^{\circ} \times 1.875^{\circ}$ |
| ERA-Interim Reanalysis | – | – | $0.5^{\circ} \times 0.5^{\circ}$ |
| MERRA Reanalysis | – | – | $0.5^{\circ} \times 0.65^{\circ}$ |

tence and $y_{t-1}$ is a weaker source to the target $x_t$ (this behavior is often observed in the climate system where persistence/self feedback plays an important role (Runge et al., 2014). The error bars represents two standard deviations of the 100 permuted surrogates showing the measure of uncertainty for the IE estimations. Furthermore there exists a significant positive net synergy ($\Delta I$) indicating that the two sources at higher couplings exchange synergistic information to the target even though the two

sources $y_{t-1}$ and $x_{t-1}$ are uncorrelated with each other, in other words, a certain degree of uncertainty about the system $x_t$ is reduced by knowing the state of $x_{t-1}$ and $y_{t-1}$ together. Here in this system, the synergy between the two sources ($y_{t-1}$ and $x_{t-1}$) to the prediction of target ($x_t$) might be arising from their linear combination. This shows that linear systems can exhibit synergies, which is also shown analytically in the work by Barrett (2015). The non-linear estimators, i.e., Kraskov estimator (40 k-nearest neighbors) and Kernel estimator (1.5 kernel width) also show the similar system behavior. The free parameters

i.e., kernel width (1–2 kernel widths) and number of k-nearest neighbors (20–60 neighbors) are tested and tuned for consistent and robust results.





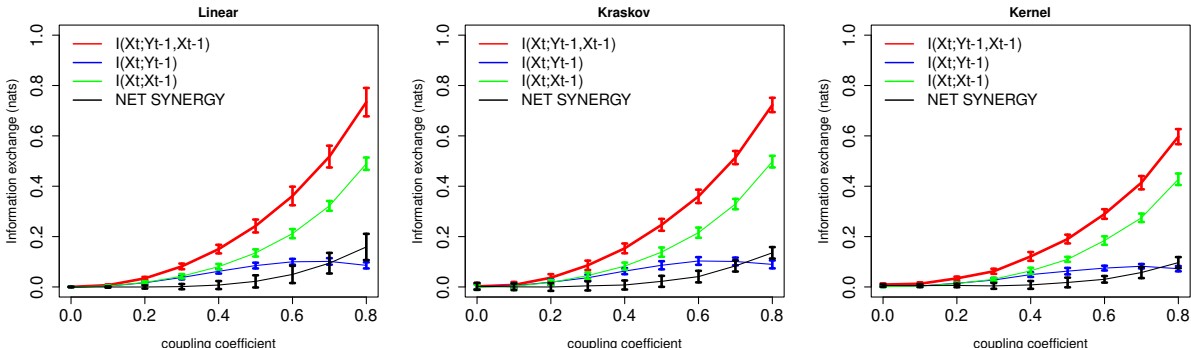

**Figure 2.** Information exchange in nats from two-source (red line), single source (green and blue lines), and net synergy (black line) to the target with Linear, Kraskov and Kernel estimators. The error bars represents two standard deviations of the 100 permuted samples.

Next, we tested another system consisting of two subsystems, coupled with each other but only having a single source as in Equation 5. From Fig. S1 (in supplementary material), the MI linear metrics shows that $I(x_t; y_{t-1}) = I(x_t; y_{t-1}, x_{t-1})$ indicating that the immediate pasts of $x_{t-1}$ does not contribute to IE for the target $x_t$. The net synergy from $y_{t-1}, x_{t-1}$ to the target $x_t$ is as expected zero. The IE from $y_{t-1}$ to $x_t$ increases as the coupling coefficient increases, which is also expected. This is also seen in Kraskov estimator (40 k-nearest neighbors) and Kernel estimator (1.5 kernel width). The free tuning parameters are tested and tuned for consistent results. Finally, among the linear systems, we tested a three-dimensional system (similar to the situation of ENSO, IOD influencing ISMR variability) with the Equation 6. Figure S2 shows that the information exchange from $I(x_t; y_{t-1}) = I(x_t; z_{t-1})$ indicating that the two sources contribute to the target system equally and moreover the IE increases with increase with coupling coefficient. This behavior is expected as observed from the governing equations. Even though the two sources are uncorrelated with each other, they exhibit positive net synergy. The similar behavior in the system is seen with non-linear Kernel estimator (1.5 kernel width) and Kraskov estimator (40 k-nearest neighbors). The free parameters are tested and tuned for consistent results.

### 4.1.2 Non-linear Heńon system

Figure 3 shows the information exchange in the non-linear Heńon system (equations given as in Equation 7). Figure 3 shows that the IE, $I(y_t; x_{t-1})$ increases as the coupling coefficient $C$ increases. It can be also observed that the opposite behavior i.e., information exchange from $I(y_t; y_{t-1})$ decreases with increasing $C$ due to the term $(1-C)y_{t-1}^2$ in Eq. 4. Also the IE from $I(y_t; y_{t-1}) > I(y_t; x_{t-1})$ indicating that the target is tightly coupled with its own past (also seen in governing equations). In this case, the two-source IE is greater than $I(y_t; y_{t-1})$ or $I(y_t; x_{t-1})$ as expected, this is because both the sources are contributing to the target future state. Here the correlation between the two sources $x_{t-1}$ and $y_{t-1}$ is almost equal to zero. The net synergy increases with increase in the coupling coefficient indicating net synergistic IE by the two-sources. For this system we used 8 number of k-nearest neighbors for Kraskov estimator and 0.5 kernel width for Kernel estimator.





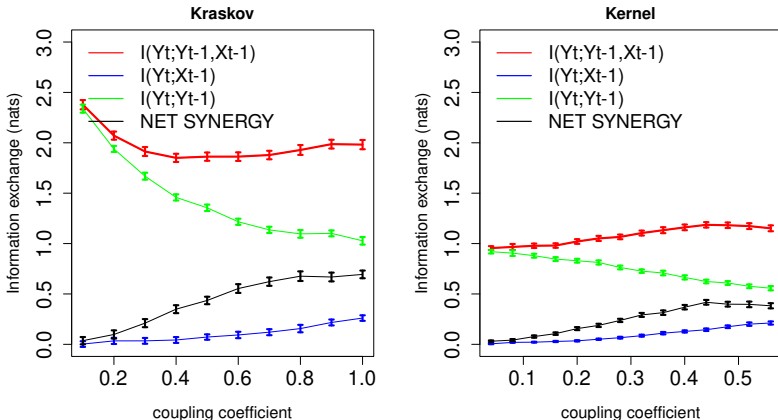

**Figure 3.** Information exchange in nats from two-source (red line), single source (green and blue lines), net synergy (black line) to target for Kraskov and Kernel estimators. The error bars represents two standard deviations of the 100 permuted samples.

The above mentioned idealized linear and non-linear examples show that some systems do exhibit positive net synergy from two-sources to target for both linear as well as non-linear systems, even when the two sources are uncorrelated. Furthermore, all the three estimators mentioned above i.e., Linear, Kernel and Kraskov estimators are able to detect consistently the two-source information exchange.

### 4.2 Application of dual-source IE to climate phenomenon

In this section, we examine the two-source IE from ENSO, IOD to the interannual variability of ISMR. Foremost, we present results obtained from the observational, reanalysis data sets and then extend our analysis of two-source IE to three GCM simulations as mentioned in Table. 1. Thereafter, we present results from our dynamically downscaled simulations with COSMO-crCLM with the three GCMs as driving models.

#### 4.2.1 Observation and reanalysis data

In the observations and reanalysis data sets, empirical orthogonal function (EOF) analysis of the detrended SST anomalies is performed over the tropical Indian ocean (25°S–20°N,50–120°E) and the tropical Pacific ocean (25°S–25°N,120°E–80°W) to obtain the major oscillations and their respective PCs. The ENSO and IOD indices are taken as the time series associated with their respective PCs obtained from the EOF spatial patterns replicating them. Figure 4 shows the second EOF patterns of the SST anomalies over the Indian ocean and first EOF patterns over the Pacific ocean for HadISST and from NCEP reanalysis. From the two SST data sets, it is observed that both ENSO and IOD like structures are captured with the second EOF and the first EOF patterns i.e., a zonal dipole like structure in the Indian ocean and the Pacific ocean respectively. We use EOF analysis





as opposed to standard indices such as the dipole mode index known as DMI (Saji et al., 1999) and Niño-3.4 to allow each

model to exhibit their own patterns as opposed to an imposed structure (Saji et al., 2006; Cai et al., 2009a, b; Liu et al., 2011).

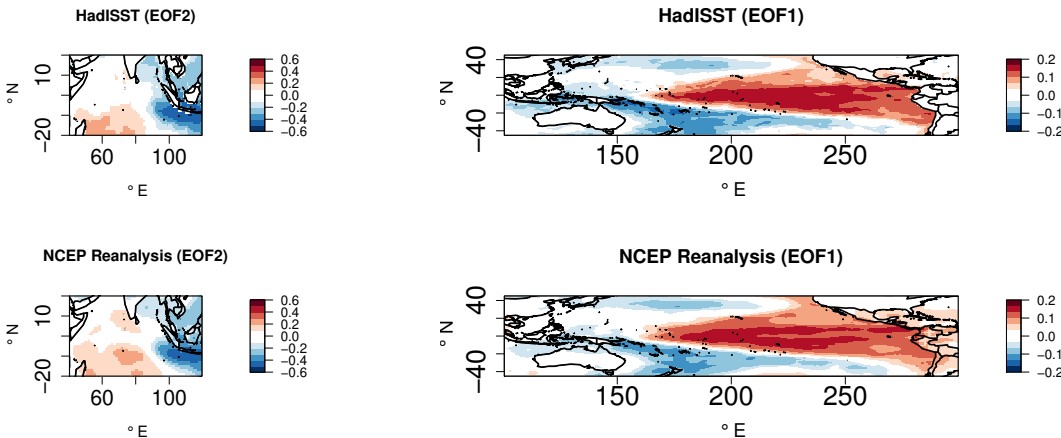

**Figure 4.** EOF2 patterns of SST anomalies (JJAS) in the Indian ocean and EOF1 patterns in the Pacific ocean for observed HadISST and
NCEP reanalysis.

To ensure that the EOF patterns in the observed SST data sets replicate the ENSO and IOD modes, the obtained PCs are
compared against the corresponding Niño 3.4 and IOD index obtained from the NOAA ESRL Physical Sciences Division,
and JAMSTEC observations (shown in Figure S3). These indices are widely used in several studies concerning the IOD and

Niño 3.4 teleconnections. The percentage of the total variance contributed by the first 20 EOFs from the Indian and Pacific
ocean SST anomalies for the seasons JJAS are also shown in Figure S3. The linear fit between the Indian ocean PCs of EOF-2
obtained from the HadISST against the observed IOD index has a correlation of about 0.78, and the correlation of NCEP
reanalysis SST with the observed IOD index is 0.77. These results are significant at a 99 % confidence level. This indicates that
the EOF2 replicates the IOD like variability for the two mentioned datasets. The percentage of the total variability contributed

by the EOF1 of the Indian ocean is about $30\%$ which is associated with the basin-scale anomalies of uniform polarity in the
Indian ocean associated with the ENSO events. The dipole mode (EOF2) explains about $15\%$ of the total variance which is
associated with the IOD. Our results for the Indian ocean EOF patterns and their respective contribution to the total variance
are consistent with the study by Saji et al. (1999). Similarly, the PCs associated with the first EOF over the Pacific ocean are
highly correlated against the observed Niño 3.4 index with a correlation value greater than 0.8 for both data sets indicating

that the EOF1 captures the ENSO like variability. The percentage of total variance contributed by the first EOF $\approx 20\%$ is also
consistent with the ENSO literature.

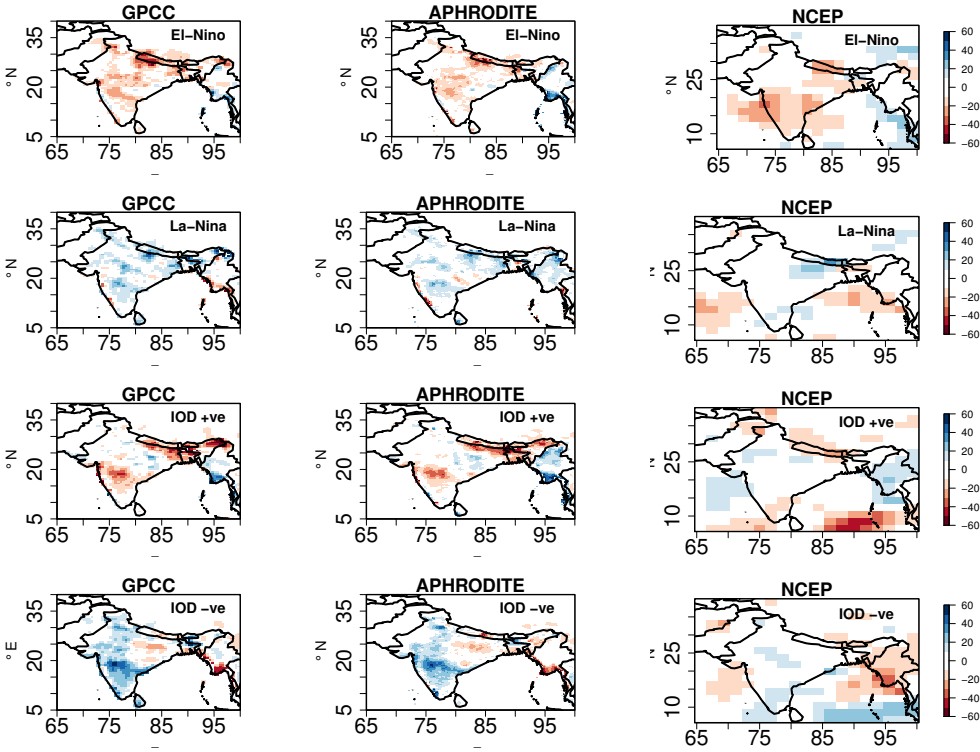

**Figure 5.** Total precipitation anomaly (mm/month) composites (JJAS) over the Indian subcontinent for El-Niño, La-Niña, positive IOD and negative IOD events observed in GPCC, APHRODITE and NCEP reanalysis data sets for the period of 1951-2005

The ENSO and IOD are known to influence the ISMR distribution across the Indian subcontinent. Hence to investigate the rainfall anomaly distribution during various phases of ENSO and IOD (i.e., El-Niño, La-Niña, IOD +ve, and IOD-ve), we plotted the anomaly composite figures ( Fig. 5 ) for the ISMR during these events. The anomalies are constructed by
subtracting the Indian subcontinent climatology mean JJAS rainfall with the rainfall months associated with various phases of IOD and ENSO. The anomaly composites with El-Niño (La-Niña) events show that most parts of Indian subcontinent receive less (more) rainfall during the El-Niño (La-Niña) phases. This behavior can be attributed to the suppression of convection over the Indian subcontinent during the El-Niño phase through the zonal and meridional circulation and vice-versa during La-Niña phase. The rainfall anomaly composites associated with the positive and negative phases of the IOD represent distinct regional
asymmetric rainfall anomalies i.e., a meridional tripolar pattern, with above than normal rainfall in central parts of India and below than normal rainfall to the north and south of it. Conversely, the negative IOD is associated with a zonal dipole having above (below) normal rainfall on the western (eastern) half of the Indian subcontinent. These results with rainfall composites during IOD phases are consistent with Behera and Ratnam (2018), where it was concluded that these rainfall anomaly patterns are due to the differences in the atmospheric responses and the associated differences in moisture transports to the region during
contrasting phases of the IOD. Hence, Fig. 5 indicates that both ENSO and IOD contribute to the interannual variability of the IMSR.



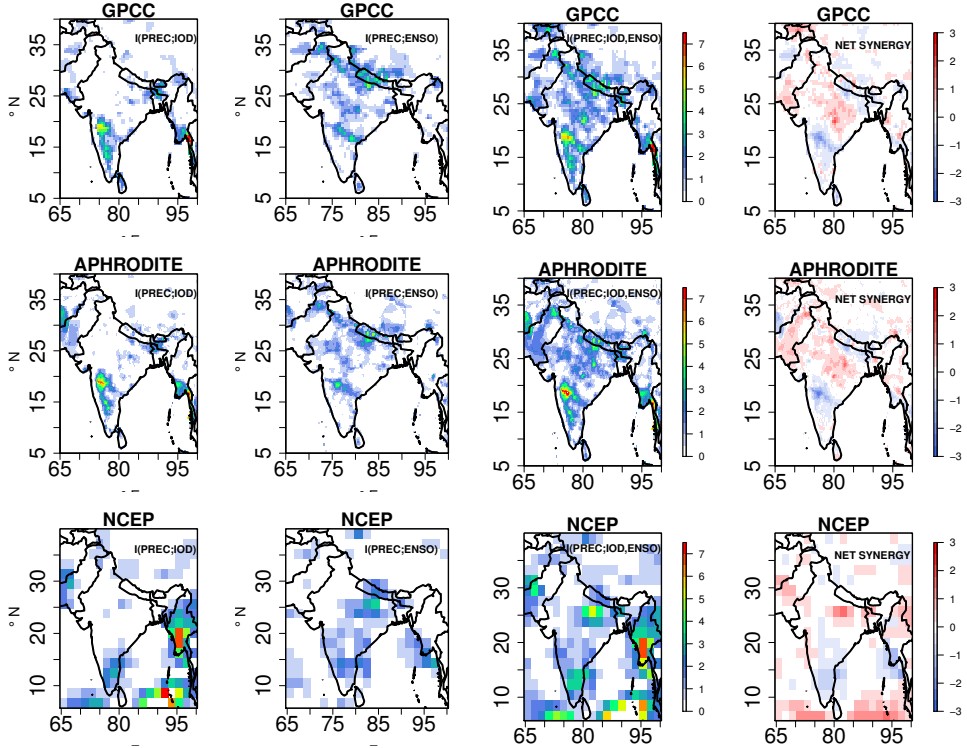

**Figure 6.** Information exchange from I(PREC;IOD), I(PREC;ENSO), two-source information exchange I(PREC; ENSO,IOD) and NET SYNERGY $\times 10^{-2}$ nats for observational data sets GPCC, APHRODITE and NCEP reanalysis. Only significant values at $95\%$ confidence intervals are plotted.

Figure 6 represents the IE from the IOD to precipitation i.e., I(PREC;IOD), ENSO to precipitation i.e., I(PREC;ENSO), the two-source IE i.e., I(PREC;IOD,ENSO) together with the NET SYNERGY for the observations GPCC, APHRODITE, and the NCEP reanalysis data sets under linear approximation. We chose various precipitation data sets to accommodate uncertainties

due to the sparse data networks, especially in regions with complex topography. The observed IE from IOD to total precipitation i.e., I(PREC;IOD) shows that the IOD transmits information to the southwest sector of the Indian subcontinent especially the lee-ward side of the western ghat regions in GPCC and APHRODITE data sets. This feature is slightly shifted to the east in the NCEP reanalysis data sets. All the IE plotted values are significant at $95\%$ confidence level obtained from 100 surrogate samples. Some regions in the northeast sector also are influenced by the IE from IOD which is replicated in all three

observational data sets. It is interesting to note that the location at which the IE from IOD to the precipitation over the Indian subcontinent matches the significant rainfall anomalies shown in Fig. 5. The I(PREC;ENSO) shows that the northern parts of the Himalayas, central India receive information from the Pacific ocean in all the three data sets, this also matches the anomaly locations shown in Fig. 5. The two-source information exchange covers most parts of the Indian subcontinent indicating that both ENSO and IOD contribute to the ISMR during JJAS seasons. Also, interestingly from the NET SYNERGY plot, a positive

net synergy over certain parts of central India also known as monsoon core region is observed, indicating that both ENSO and





IOD synergistically contribute to the interannual variability of ISMR. Furthermore, the ENSO and IOD share net redundant information (negative net synergy) in the southern sector of the Indian sub-continent. The Kraskov estimator (Fig. S4) and the Kernel estimator (figure not shown) also show similar IE patterns over the Indian subcontinent with 30 k-nearest neighbors for Kraskov and 1.5 kernel width for Kernel estimators. In addition, we also checked the two-source IE patterns in the two reanalysis datasets, MERRA and ERA-Interim (1980-2005), shown in Fig. S5 and Fig. S6. It is found that in both the data sets, similar IE patterns are replicated i.e., positive net synergy in central India and net redundant information in southern part of the Indian subcontinent.

The net synergy between the ENSO and IOD to the ISMR interannual variability indicates that the central India monsoon rainfall predictability lies in knowing the states of ENSO and IOD together than by knowing the states of ENSO and IOD individually (similar to the idealized test case of example 3). This is also exactly similar to the XOR logic gate, where the uncertainty of the output is known only with the simultaneous knowledge of the two input states. To understand the information synergy physically, we show the moisture transport figures from the NCEP reanalysis datasets for various phases of ENSO and IOD during the JJAS. From Fig. 7 it is observed that the anomalous negative moisture flux during the El-Niño is compensated with the positive moisture flux anomaly by IOD +ve especially in central India, and vice-versa during the La-Niña and IOD-ve events. It is known that the El-Niño events are often associated with IOD+ve events (Behera and Ratnam, 2018) and vice versa (the ENSO and IOD are positively correlated in our data sets). From the precipitation composites (Fig.4), in central India, an anomalous negative (positive) rainfall during the El Niño (La Niña) is observed, and during the IOD+ve (IOD-ve) a positive (partly negative) anomalous rainfall is observed. This could explain why both the IOD and ENSO states should be known together to explain the variability of the central Indian subcontinent rainfall as the IOD and ENSO are having compensating effects. This compensating behavior is not seen in the southern or northern part of the Indian subcontinent, hence this could explain the net redundant information between ENSO and IOD to the precipitation to the southern region. The readers are referred to Fig.3 by Barrett (2015) to further explore the relation of synergy dependence on the compensating influence from both sources, i.e., the correlation between two sources and to their targets respectively.





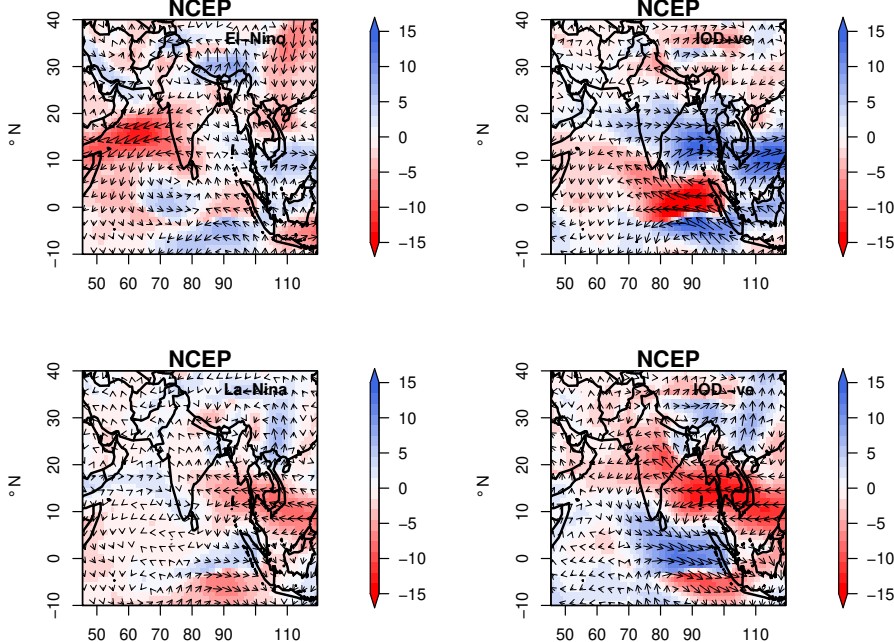

**Figure 7.** Moisture flux anomalies (g/kg m/sec) over the Indian subcontinent (JJAS) for El-Niño, La-Niña, IOD+ve and IOD-ve events observed in NCEP reanalysis data sets for the period of 1951-2005.

### 4.2.2 Global and regional climate model simulations

Next, we are performing the same analysis, starting with the EOF patterns from the SST fields obtained from the three GCM simulations listed in Table 1, to investigate how the ENSO and IOD associated variability in the Indian and Pacific oceans are represented. Figure 8 shows the second EOF pattern of the SST anomalies over the Indian ocean and the first EOF pattern in the Pacific ocean, for the GCM simulations of MPI-ESM-LR, Nor-ESM-M, and EC-EARTH. It is found that all the GCM simulations replicate the zonal dipole like patterns over the Indian ocean and Pacific ocean similarly as the observations. The

percentage of the total variability contributed by EOF1 of the Indian ocean is about 30% in all the GCM simulations (Fig. S7) which is comparable to the observations. The EOF2, which is associated with the IOD, explains about 15% of the total variance in all the GCMs, also similar to observations. The percentage of total variance contributed by the first EOF is between 20 − 25% in all the GCM simulations in the Pacific ocean, which is similar to variance in the observations. Thus, these results indicate that the variability associated with the SST anomalies over the Indian and the Pacific ocean is represented in the

three GCM simulations. The SST anomaly composites during various phases of IOD and ENSO events (Fig. S8 and Fig. S9) show that most of the GCM simulations can replicate the SST anomaly composite patterns found during the IOD+ve events in HadISST (Fig. S8). On the contrary, during IOD-ve events, the MPI-ESM-LR portrays unrealistic warm anomalies throughout the Indian ocean. Over the Pacific ocean, the MPI-ESM-LR and EC-EARTH have an unrealistic westward extension of the warm (cold) pool during El Niño (La Niña) events. The patterns from Nor-ESM-M are closer to the observation, shown in Fig.





S9. The unrealistic westward extension of the SSTs in EC-EARTH and MPI-ESM-LR simulations might influence the walker circulation through unrealistic large scale teleconnections patterns.

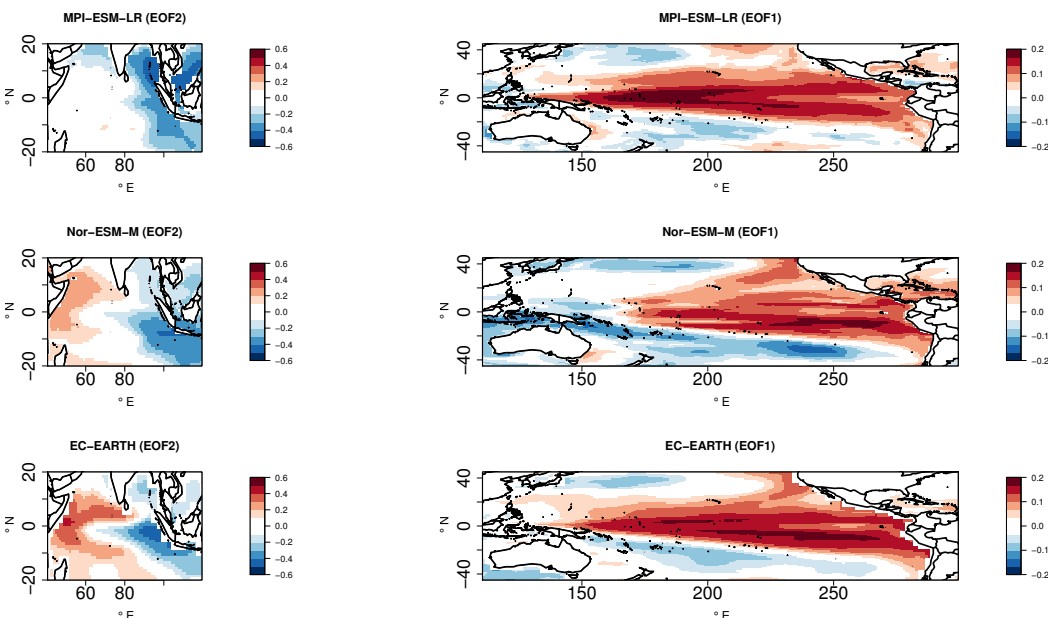

**Figure 8.** EOF2 patterns of SST anomalies for (JJAS) in the Indian ocean and EOF1 patterns for (JJAS) in the Pacific Ocean for three GCM simulations, i.e., MPI-ESM-LR, Nor-ESM-M and EC-EARTH for the period of 1951-2005.

Figure 9 represents the ISMR anomaly composites during the El-Niño, La-Niña, IOD+ve and, IOD-ve events for the three GCM simulations, the MPI-ESM-LR, Nor-ESM-M, and EC-EARTH, when selecting the associated years given by the respective PCs. The rainfall anomaly composites associated with the positive phase of ENSO show dry conditions over the
northern/northwest parts of the Indian subcontinent in the MPI-ESM-LR, dry conditions throughout the Indian sub-continent in Nor-ESM-M. The EC-EARTH simulation does not show a clear rainfall anomaly signal. Similar opposite polarity of rainfall anomalies are observed in the La-Niña conditions in the MPI-ESM-LR and Nor-ESM-M simulations, while slight wet conditions in north-east India in EC-EARTH. For the IOD+ve events, MPI-ESM-LR shows dry conditions in the southwest, while the Nor-ESM-M simulation shows dry conditions in the northwest and the Himalayan region, the EC-EARTH does not show
any variability. The Nor-ESM-M during the IOD-ve phase shows overall positive anomaly, while no clear signal is observed in MPI-ESM-LR and EC-EARTH. Overall, the ENSO phase signal is better replicated in Nor-ESM-M simulation and partly in MPI-ESM-LR as in the observations, while most of the GCM simulations failed to replicate the regional rainfall asymmetric response in IOD events as in observations (except Nor-ESM-M, which partly can replicate the dipole patterns). This might be due to the coarse resolution of GCMs which may not be able to replicate the fine-scale precipitation response to the IOD.

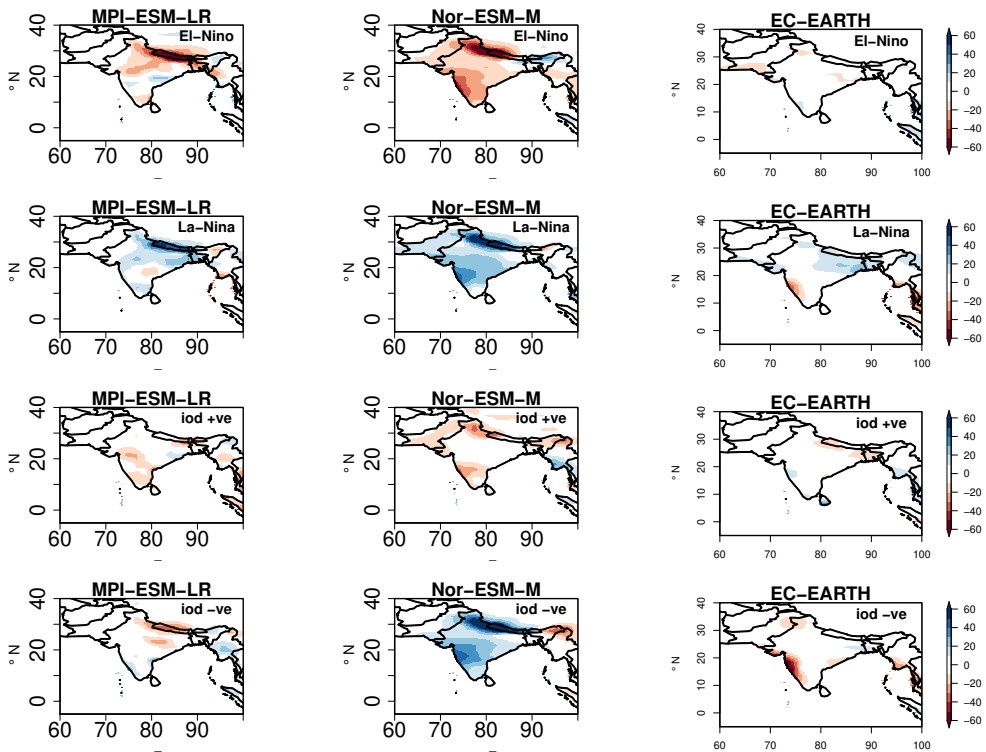

**Figure 9.** Total precipitation anomaly composites over the Indian subcontinent (JJAS) for El-Niño, La-Niña, positive IOD and negative IOD events in MPI-ESM-LR, Nor-ESM and EC-EARTH simulations(1951-2005)

Figure 10 represents the IE spatial patterns from the IOD and ENSO i.e., I(PREC;IOD), I(PREC;ENSO), the two-source IE, I(PREC;IOD,ENSO) together with the NET SYNERGY over the Indian subcontinent in the three GCM simulations i.e., MPI-ESM-LR, Nor-ESM-M, and EC-EARTH with the linear estimator. The information exchange from IOD to total precipitation in MPI-ESM-LR shows that the information from the IOD is exchanged to the southeastern part of the Indian Subcontinent. This is contrary to what is seen in the results from the observations, where most of the IE takes place to the leeward side of the

western ghats and the northeastern sector of India. The Nor-ESM-M simulation shows that IE from IOD is transmitted to the western side of the Indian subcontinent, where the observed significant anomalies are noted in Fig. 10. The EC-EARTH does not show any information exchange from IOD to the land points over the Indian sub-continent. The IE(PREC;ENSO) show that the northern parts of the Himalayas and north west-central India receive information from the Pacific ocean in MPI-ESM-LR. For Nor-ESM-M, the western ghats and its leeward side are influenced by ENSO. The EC-EARTH does not show as much

IE as the Nor-ESM-M or MPI-ESM-LR over the Indian continent, with an exception for some scattered locations over the Himalayas.

The two-source information exchange IE(PREC;ENSO,IOD) covers the northwest part of the Indian subcontinent for MPI-ESM-LR and the extreme southeast. For Nor-ESM-M the information exchange covers mostly the western part of India. The EC-EARTH show IE over isolated places of northeast India. These results indicate that the three GCMs exhibit a IE





pattern which is different from the observed patterns. Moreover, the results of the NET SYNERGY show that MPI-ESM-LR
does not show any net synergistic IE over the Indian subcontinent, while in Nor-ESM-M the IOD and ENSO share common
information over the west of India. EC-EARTH show less net synergy over the Indian sub-continent. Overall, the results from
the IE exchange differ from the observations, seen for all the three GCM simulations. These results are consistent with Kernel
and Kraskov estimators (figures not shown).

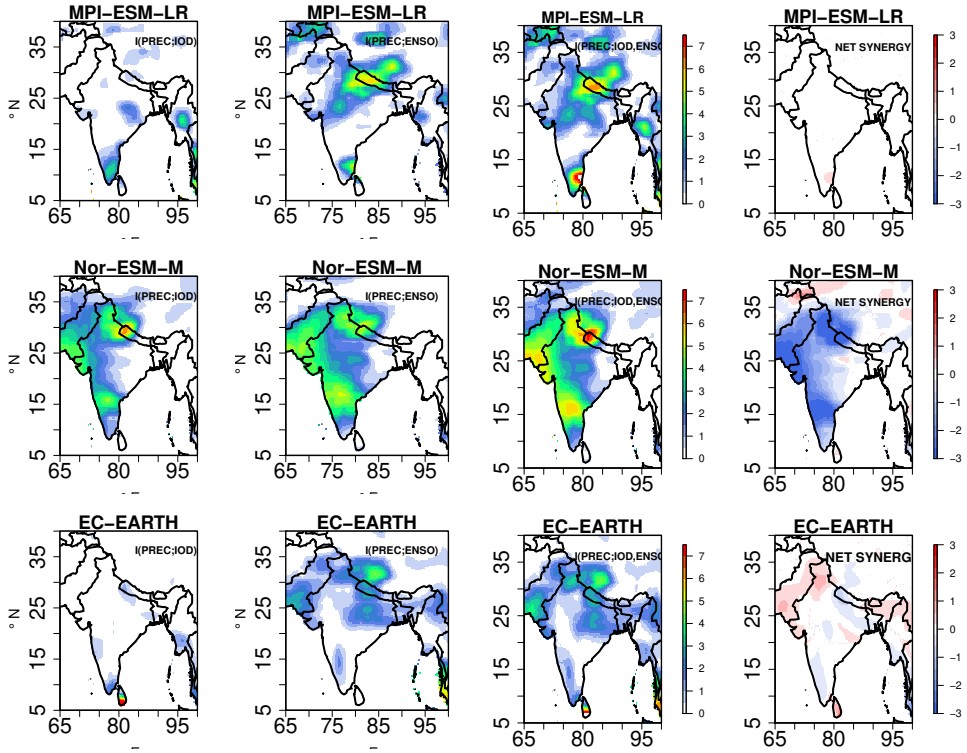

**Figure 10.** Information exchange from I(PREC;IOD), I(PREC;ENSO), two-source information exchange I(PREC; ENSO,IOD) and NET
SYNERGY $\times 10^{-2}$ nats for the GCM simulations MPI-ESM-LR, Nor-ESM-M and EC-EARTH for JJAS (1951-2005). Only significant
values at 95% confidence intervals are plotted.

Next, we are investigating how the two-source information exchange is represented when we dynamically downscale the
three GCM simulations (MPI-ESM-LR, Nor-ESM-M, and EC-EARTH) with the regional model COSMO-crCLIM (0.22°). We
are applying the same two-source information exchange method on the RCM fields as we have done for the GCM simulations.
However, since the RCM simulations are only covering a limited area, namely the South Asian CORDEX domain, we had
to combine the RCM results with the GCM simulations, in particular for the EOF-analysis over Indian and Pacific oceans.
Figure 11 represents the ISMR anomaly composites during the positive IOD+ve, IOD-ve, El-Niño, and La-Niña events for
the COSMO-crCLM RCM simulation driven with three GCM simulations, the MPI-ESM-LR, Nor-ESM-M, and EC-EARTH.
Here we are selecting the same years as given by the principal components from the driving GCM simulations. The rainfall
anomaly composites associated with the El-Niño events show dry conditions over the northern parts of Himalayas for the





downscaled MPI-ESM-LR and wet conditions in western ghats and isolated parts in central India. During the La-Niña phase,

dry conditions in the central Indian subcontinent, western ghats and wet conditions elsewhere are observed. In the downscaled

Nor-ESM-M, dry (wet) signal is observed throughout Indian subcontinent during El-Niño (La-Niña) phases. In the downscaled

EC-EARTH, dry regions are noted throughout most parts of Indian subcontinent during El-Niño, while dry conditions are seen

in central India and wet conditions elsewhere in La-Niña phase. The rainfall anomalies composites associated with the positive

IOD in the observations, i.e., a meridional tripolar pattern with above than normal rainfall in central parts of India and below

than normal rainfall to north and south of it is only observed in the downscaled Nor-ESM-M. Similarly, the negative IOD in

downscaled Nor-ESM-M is associated with a zonal dipole having above (below) normal rainfall on the western (eastern) half

of India similar to that of the observations as seen in Figure 6. Overall, these results suggest that the downscaled results from

Nor-ESM-M better reproduces the spatial patterns of precipitation anomalies associated with ENSO and IOD, when comparing

to the observations, than the downscaled results from EC-EARTH and MPI-ESM-LR.

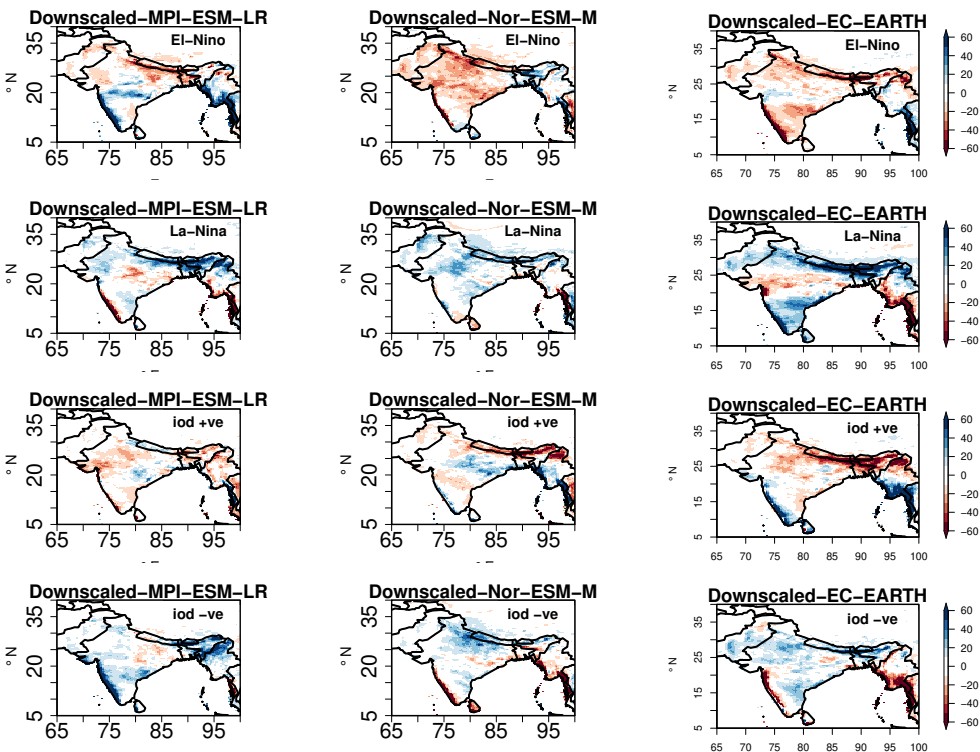

**Figure 11.** Total precipitation anomaly composites over the Indian subcontinent for El-Niño, La-Niña, positive IOD and negative IOD events for the downscaled COSMO-crCLM simulations driven by MPI-ESM-LR, Nor-ESM-M and EC-EARTH GCM simulations for JJAS (1951-2005)

Figure 12 represents the IE patterns over the Indian subcontinent for the downscaled RCM simulations with the Linear

estimator (these patterns are also consistent with Kraskov and Kernel estimators). The net synergy in central India, and shared

information in southern India is better represented in the downscaled Nor-ESM-M simulation, compared to the downscaled





MPI-ESM-LR and downscaled EC-EARTH. This is in agreement with the results from the GCM simulation, where it was found that Nor-ESM-M simulation had a better replication of ENSO and IOD induced anomalous precipitation structures than the two other GCMs (see Fig. 11). These results are interesting, even though all the COSMO-crCLM simulations have the same physics and dynamics, only downscaled Nor-ESM-M replicated realistic patterns of IE. The improvement in results in downscaled Nor-ESM-M can be attributed to a more realistic large-scale information coming from the GCM simulation, such as the moisture flux transport during various phases of ENSO and IOD events (see Fig. S10 – Fig. S14 and Fig.7). For the MPI-ESM-LR and EC-EARTH GCM simulations, the moisture flux anomalies are very different from the reanalysis fluxes and thus seem misrepresented. A better replication of the moisture flux anomaly in Nor-ESM-M GCM simulation during ENSO and IOD might be from a better simulation of the large scale circulation patterns, like the Walker and Hadley circulations, due to the better representation of the SST than the two other GCM simulations (Fig. S8 and Fig. S9). The RCM simulation results exhibit similar moisture flux anomalies compared to the driving GCM simulations, in which the downscaled Nor-ESM-M outperforms the downscaled MPI-ESM-LR and downscaled EC-EARTH. These results indicate that a realistic large-scale signal from the GCM simulations (e.g., the moisture transport and SST anomalies) is essential for an RCM to properly improve the GCM results in terms of IMSR variability. When the large-scale signal from the GCM is incorrect, and wrong moisture fluxes are imposed on the lateral boundaries of the RCM, the downscaled results are hampered.

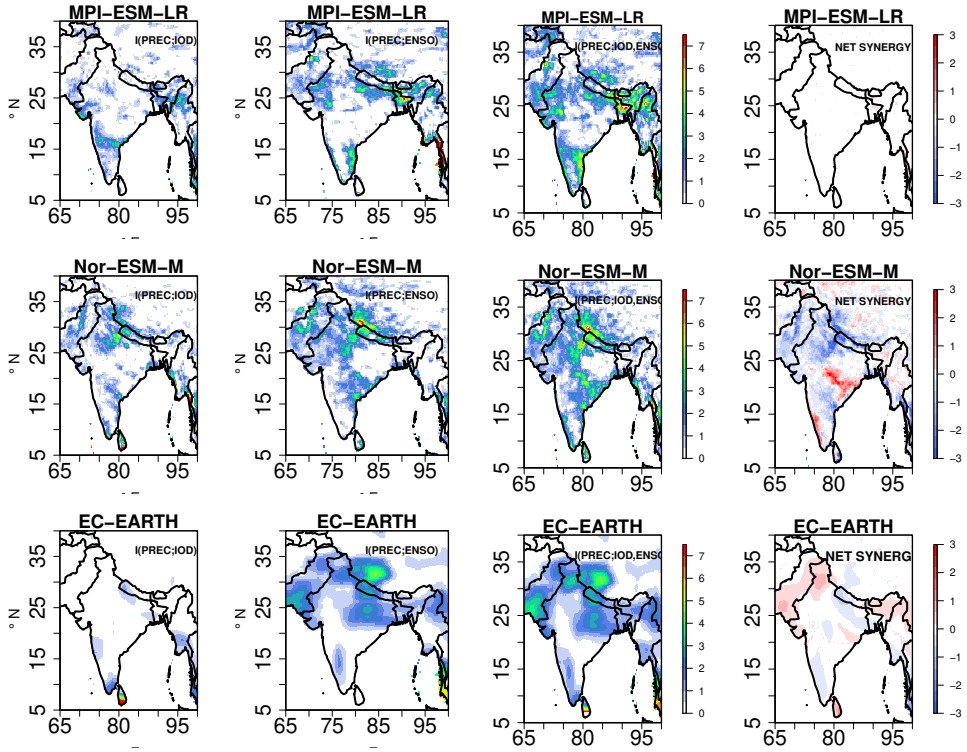

**Figure 12.** Information exchange from I(PREC;IOD), I(PREC;ENSO) and two- source information exchange I(PREC; ENSO,IOD), NET SYNERGY $\times 10^{-2}$ nats for the downscaled COSMO-crCLM simulations for JJAS (1951-2005). Only significant values at 95% confidence intervals are plotted.

## 5    Conclusions

In this article, we explored two-source information exchange (IE) from ENSO and IOD (quantified by SST variabilities in the
Pacific and Indian oceans) to the Indian Monsoon Summer Rainfall (IMSR) interannual variability. But, first, we used simple idealized linear and non-linear dynamical systems to demonstrate the concepts of two-source IE. Results showed that both the linear and the non-linear idealized systems can exhibit positive net synergy (i.e., the combined influence of two sources is greater than their individual contributions). Interestingly, two uncorrelated sources can show positive net synergistic IE to a target.

The two-source ENSO and IOD to IMSR IE was explored in observations, reanalysis data sets, and in three GCM simulations which were also further dynamically downscaled with the RCM. The results from the observations and reanalysis data suggest that both IOD and ENSO influence the interannual variability of the ISMR throughout most parts of Indian subcontinent. Interestingly, we found that IOD and ENSO exhibit positive net synergy over central India, which is the monsoon core region, and net redundant information over the southern part of India.

The IE patterns in the three GCM simulations differ from that in the observations. However, the GCM Nor-ESM-M better captured the precipitation anomalies from ENSO and partly from IOD than the other two GCMs. Previous studies also showed that Nor-ESM-M outperforms other CMIP5 GCM simulations in terms of rainfall climatology, and most aspects of the climatological annual cycle and interannual variability in the Indian subcontinent (Sperber et al., 2012; McSweeney et al., 2015).

Downscaling Nor-ESM-M simulation with the RCM COSMO-crCLM better replicated the observed IE patterns than downscaling the MPI-ESM-LR and EC-EARTH simulations. Importantly, the downscaled Nor-ESM-M IE results are in better agreement with the observations than the Nor-ESM-M results. Downscaling Nor-ESM-M adds value to the GCM simulation. This can not be concluded here for downscaling of MPI-ESM-LR and EC-EARTH simulations. Downscaling the latter simulations did not add value because of a missing realism in their large-scale SST patterns and horizontal moisture flux variability, which are important RCM boundary conditions and which were better represented in the Nor-ESM-M simulation. Downscaling did not compensate errors in the large-scale driving simulations. These results highlight the importance of the choice of GCM simulations when performing dynamically downscaling for high-resolution regional climate projections.

Finally, we propose to use the two-source IE metric as a complementary tool to gain additional insight into the climate system and to perform process-oriented climate model evaluation.

*Code availability.*   The analysis is done in R and all codes can be provided upon request.

*Data availability.*   The GCM and the RCM datasets are available at https://esgf-data.dkrz.de/projects/esgf-dkrz/. The ENSO and IOD index are taken from http://www.esrl.noaa.gov/psd and http://www.jamstec.go.jp/ respectively.

*Author contributions.*   The concept was proposed by, B.A. Funding was acquired by B.A. The information theory algorithms were developed by, P.K.P. and C.P. The RCM simulations were performed by, S.S. with assistance from P.K.P. The manuscript was written by, P.K.P. and
reviewed by, B.A., S.S., and, C.P. All authors have read and approved the final manuscript.

*Competing interests.*   The authors declare that they have no conflict of interest.

*Acknowledgements.*   The authors acknowledge the support by the German Research Foundation ("Deutsche Forschungsgemeinschaft", DFG) in terms of the research group FOR 2416 "Space-Time Dynamics of Extreme Floods (SPATE)". The authors also thank Joseph T Lizer for providing the JDIT open source toolkit. CLMcom-ETH-COSMO-crCLIM-v1-1 simulations were run on Piz Daint at CSCS (Switzerland),
and we acknowledge PRACE for awarding us access and computing time to Piz Daint.



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
