# Peer review of "The synergistic impact of ENSO and IOD on the Indian Summer Monsoon Rainfall in observations and climate simulations - an information theory perspective"

_Earth System Dynamics, 2020_

## Short Comment (SC1) · 15 Jul 2020

Since it has become obvious that common-mode tidal forcings control the majority of climate indices, as a first step one should consider how the tidal factors play into the models. See attached figures for AMO and ENSO. Once this causality is understood, then it will be much easier to deal with other interactions. ENSO and IOD have just slight variations on the tidal forcing. Cheers

[Figure]

**Fig. 1.**

[Figure]

**Fig. 2.**

[Figure]

**Fig. 3.**

---

## Referee Comment (RC1) · Benjamin L. Ruddell (Referee) · 24 Jul 2020

Owing to the intense workloads created by the COVID-19 crisis, my review will be brief and will stick to the high level.

Overall, this is a very interesting manuscript and deserves to be published with minor revisions. The technical methods are sound at the level of detail I am able to review them. The possible improvements lie in the communication and interpretation of the results.

I suggest shortening the manuscript dramatically, especially by moving to a supplement the early portions of the results and methods where the authors prove that the metrics capture the kind of information content and synergy that is relevant to this climate coupling. We already know these metrics work, so your validation is important as due diligence but not as an important result of the paper, in my opinion.

I suggest considering and including more concepts and language about coupling, and/or causation (coupling is better in my opinion), as opposed to information content and synergy. Coupling, where appropriately interpretable, is a more intuitive and useful concept that is much more broadly understood than synergy or information content, and is better communication. I believe you are talking about a physical coupling between oceanic processes and the monsoon here, at least in part. Some of my papers get into process coupling concepts and language, including the original Ruddell and Kumar 2009 in Water Resources Research.

The major change I'd like to see is the inclusion of more interpretation of these results in terms of physical atmospheric process dynamics. What does this information content and synergy mean, physically? Can you confirm or reject a hypothesis about the processes that are are causing it, using these information statistics? What does this mean? Actually testing a hypothesis would be the best, but more discussion in the conclusions is also very helpful.

I suggest reviewing and possibly including additional work from Bookhagen, Knuth, Brunsell, and Kurths, who have all published on spatially gridded information content and flows; Kurths' work on climate networks may be particularly useful here as a comparison or corroboration because it touches directly on the parts of the world you're studying.

https://scholar.google.com/citations?user=_UuIymgAAAAJ&hl=en&oi=ao

https://scholar.google.com/citations?user=ZcY4VC8AAAAJ&hl=en&oi=sra
https://scholar.google.com/citations?user=cluDFKcAAAAJ&hl=en&oi=ao

https://scholar.google.com/citations?user=iwzqdyQAAAAJ&hl=en&oi=ao

---

## Referee Comment (RC2) · Didier Vega-Oliveros (Referee) · 17 Aug 2020

In this work, the authors presented a new method for understanding the synergistic, unique, and redundant information exchange from the ENSO and IOD phenomena on the Indian Summer Monsoon rainfall. For this propose, they employed some tools from information theory, the mutual information, and entropy to estimate how two other sources can estimate a third variable, quantifying the unique information, redundant information, and synergistic information contributions. The authors considered three estimation techniques and showed how well their approach works and makes sense in

three linear and one non-linear artificial systems. After that, they used several sources of real data from observational and reanalysis data sets. They showed that their technique obtains similar conclusions about ENSO and IOD's synergistic impact on the Indian Summer Monsoon Rainfall. Further, they evaluated their initial conclusions from observational data in global and regional climate models.

The work has its merits, is interesting and relevant for the area, with the potential of future works and interdisciplinary developments. The manuscript is clear and easy to follow, but quite extensive, in which the structure and order of the sections could be improved. First, in the Introduction, the sentences: "Shannon (1948) first introduced the concept of information entropy, which quantifies the average uncertainty of a given random variable. The IE between two subsystems X and Y can be understood as the average uncertainty reduction about X in knowing Y or vice versa." and all the part of "The IE in a system composed of two-source systems Y and Z ... alone but by jointly knowing their states together." are adequate to the Material and Methods section than the Introduction. Please, consider moving these parts to the method Section and refer to it in the Introduction if necessary. This reviewer also suggests moving the Material and Method sections to be the last part of the manuscript and promptly presenting the results of the work. Besides, the authors can move some broadly and detailed concepts to the supplemental material.

Before recommending the article's acceptance, there is a further analysis that this reviewer will ask the authors. Could you please run the same analysis in observational data for the same regions but instead considering an outside temporal season (e.g., DJFM)? In this way, we can get more insides and understanding of the proposed method and how well are the behaviors and results. For example, if one wants to check other regions and phenomena, to discover new dynamical/physical connections, it would be feasible to apply this method and found if there are pieces of evidence of physical connection or not, (like can be done with many other approaches and knowing their drawbacks). With a negative test in real data, the authors can show the robustness of their method and the ability to be used to test other systems.

Minor comments:

"provides an lowerbound for..." => "provides a lowerbound for..."

In lines 417 and 422, is it IE(...) or should be I(...)?

In lines 358 and 429, why the authors did not include these figures in the Supplemental material?

About the estimator K-nearest neighbors (called here as Kraskov), how was the approach employed to find the best k by the authors? Did they try all possible values and choose the best one? How did they define or evaluate the best k?

In terms of code availability, it is a big plus and highly recommended that the authors publicly available their code in open source platforms (like GitHub, for instance). Therefore, other scholars and the community can use it to replicate the conclusions and compare it with their methods in future works.

---

## Author Comment (AC1) · 7 Sep 2020

Dear Dr Didier Vega-Oliveros and Dr Benjamin L. Ruddell,

Thank you very much for your very helpful and constructive comments. Here, you can find point by point replies to your comments and suggestions.

Reviewer comments in: Black

Our reply in: Blue

**Review by Dr Didier Vega-Oliveros:**

1. The work has its merits, is interesting and relevant for the area, with the potential of future works and interdisciplinary developments. The manuscript is clear and easy to follow, but quite extensive, in which the structure and order of the sections could be improved.

   Thank you for your nice comment. We hope that our application of information theory methods on climate data will open up new perspectives in the climate science community.

   We agree that our manuscript is a bit extensive. However, as our work applies various methods from information theory to climate data, we had to discuss our methods in detail and test them before applying to observations, reanalysis, GCMs, and RCMs for the benefit of readers who are new to information theory. Nevertheless, following your suggestions, we moved few sentences about information theory from Introduction to Methodology Section and also some idealized test cases to Appendix Section as suggested by Dr Benjamin Ruddell (details of the changes are provided in our replies to your specific comments).

2. First, in the Introduction, the sentences: "Shannon (1948) first introduced the concept of information entropy, which quantifies the average uncertainty of a given random variable. The IE between two subsystems X and Y can be understood as the average uncertainty reduction about X in knowing Y or vice versa." and all the part of "The IE in a system composed of two-source systems Y and Z ... alone but by jointly knowing their states together." are adequate to the Material and Methods section than the Introduction. Please, consider moving these parts to the method Section and refer to it in the Introduction if necessary.

   We moved the lines you stated to the Methodology Section 2 (changes are highlighted with blue/red color in edited manuscript attached below for reviewers quick reference). However, we had retained some important brief explanations of information theory in the Introduction Section for the readers to have a quick understanding of the information exchange concepts.

3. This reviewer also suggests moving the Material and Method sections to be the last part of the manuscript and promptly presenting the results of the work. Besides, the authors can move some broadly and detailed concepts to the supplemental material.

   Thank you for the suggestion. We would like to start with the Methodology Section as it helps the reader to follow the new metrics from information theory before reading the results from the idealized as well as the climate applications. Hence, we maintained the same order in our revised manuscript.

4. Before recommending the article's acceptance, there is a further analysis that this reviewer will ask the authors. Could you please run the same analysis in observational

data for the same regions but instead considering an outside temporal season (e.g., DJFM)? In this way, we can get more insides and understanding of the proposed method and how well are the behaviors and results. For example, if one wants to check other regions and phenomena, to discover new dynamical/physical connections, it would be feasible to apply this method and found if there are pieces of evidence of physical connection or not, (like can be done with many other approaches and knowing their drawbacks). With a negative test in real data, the authors can show the robustness of their method and the ability to be used to test other systems.

We agree with your point of concern on testing the methods outside the seasons (DJFM) where the IOD and ENSO combined influence is not observed. As per your suggestion, we have tested our methods over the observational data for the months of DJFM and as expected, the ENSO and IOD do not synergistically contribute to the rainfall for the months of DJFM (we added this line in our revised manuscript). Here are our results and detailed discussion.

Figure R1 represents the EOF modes for ENSO and IOD over the Pacific and Indian Oceans respectively. From the literature, it is known that the ENSO mode over the Pacific Ocean peaks at the end of the year [1], while the IOD ends in November and starts again in May[2]. From the figures R1-R3, it is seen that the ENSO SST anomalies are clearly formed. The IOD SST anomalies for the months of DJFM are not well formed as compared to the anomaly structure seen for the month of JJAS (Fig. 3 in our manuscript). These results fit well with the existing literature.

[Figure]

**Fig. R1**: EOF2 patterns of SST anomalies (DJFM) in the Indian ocean and EOF1 patterns in the Pacific ocean for observed HadISST and NCEP reanalysis.

[Figure]

**Fig. R2**: SST composites for various phases of IOD and ENSO events for the months of DJFM for HadISST observational data.

[Figure]

**Fig. R3**: SST composites for various phases of IOD and ENSO events for the months of DJFM for NCEP Reanalysis data.

Figure R4 shows that the linear fit between the Indian Ocean PCs of EOF-2 obtained from the HadISST against the observed IOD index has a correlation of about 0.51, and the correlation of NCEP reanalysis SST with the observed IOD index is 0.12. The NCEP reanalysis data for the months of DJFM is unable to replicate the IOD structures as compared to the observed HadISST. However, the correlation of both data sets with the IOD index is higher in JJAS compared with the DJFM. This can be attributed to the weak amplitude of IOD during DJFM seasons compared to JJAS. The PCs associated with the first EOF over the Pacific Ocean are highly correlated against the observed Niño 3.4 index with a correlation value of 0.86 for both data sets indicating that the EOF1 captures the ENSO like variability. The correlation for the months of DJFM is greater than the JJAS months. This might be due to the ENSO peak in the months of DJFM. The variability of the IOD and ENSO modes for the months of DJFM is also consistent with the literature[3].

[Figure]

**Fig. R4**: Regressions of PCs obtained from their respective EOFs over the Indian and Pacific Oceans with the observed IOD and Nino 3.4 Index and their associated percentage contribution to the total variance for HadISST and NCEP reanalysis SST data sets for the months of DJFM

[Figure]

**Fig. R5**: Total precipitation anomaly (mm/month) composites (DJFM) over the Indian subcontinent for El-Niño, La-Niña, positive IOD and negative IOD events observed in GPCC, APHRODITE and NCEP reanalysis data sets for the period of 1951-2005.

Figure R5 represents the anomalies constructed by subtracting the Indian subcontinent climatology mean DJFM rainfall with the rainfall months associated with various phases of IOD and ENSO. The anomaly composites with El-Niño (La-Niña) events show that ENSO events influence the precipitation in winter over the northwest India. This influence is attributed to the intensification of the western disturbances over the northwest India due to the baroclinic response[4]. The composites of precipitation anomalies during the IOD[+ve] events show more than normal rainfall near the east coast and central India for the DJFM seasons for GPCC and APHRODITE datasets. The Influence of IOD on the winter rainfall over India is less studied compared to the JJAS rainfall. However, among the limited studies Kripalani and Kumar[5] 2004, showed the influence of IOD on the North East Monsoon rainfall during the months of October , November and December with IOD[+ve] events leading to more rainfall over Southern India and also towards North (see Fig. 9 in Kripalani and Kumar, 2004[5]). The IOD[+ve] influence also show positive rainfall anomaly in our study and this needs a detailed investigation on the process leading to it. It is worth to note here that the rainfall during DJFM is less than rainfall amount in JJAS shown in the manuscript.

[Figure]

**Fig. R6**: Information exchange from I(PREC;IOD), I(PREC;ENSO), two-source information exchange I(PREC; ENSO,IOD) and NET SYNERGY $\times 10^{-2}$ nats for observational data sets GPCC, APHRODITE and NCEP reanalysis. Only significant values at 95% confidence intervals are plotted.

Figure R6 shows the information exchange (IE) from the IOD to precipitation i.e., I(PREC;IOD), ENSO to precipitation i.e., I(PREC;ENSO), the two-source IE i.e., I(PREC;IOD,ENSO) together with the NET SYNERGY for the observations GPCC, APHRODITE, and the NCEP reanalysis data sets under linear approximation. The observed IE from IOD to total precipitation i.e., I(PREC;IOD) shows that the IOD transmits information to the north-central sector of the Indian subcontinent in the GPCC and APHRODITE data sets. The location at which the IE from IOD to the precipitation over the Indian subcontinent matches the significant rainfall anomalies. This is also true in the case of ENSO, where it influences the north-west sector of India. The net synergy plot show that the IOD and ENSO do not share any net synergistic information over the subcontinent. This is expected as the IOD and ENSO are known to act mutually in JJAS than the DJFM season. Our results reiterate the same here.

Minor comments/Suggestions

5.  "provides an lowerbound for..." => "provides a lowerbound for..."

    Rephrased as per the suggestion.

6.  In lines 417 and 422, is it IE(...) or should be I(...)?

    Thank you, as you mentioned it should be I(..). We changed it in the manuscript.

7.  In lines 358 and 429, why the authors did not include these figures in the Supplemental material?

We tested the Linear, Kraskov and Kernel estimators for the idealized and climate applications. Since our manuscript is quite extensive and furthermore the climate application is near Gaussian, we showed the Linear estimator in the manuscript (the linear estimator is robust than the non-linear Kraskov or Kernel estimator as the non-linear estimators depend on the free tuning parameters in the estimation of PDF (Pothapakula et al., 2019)). But now we have included additional figures in the supplementary material (Fig.S4, Fig.S5, FigS11, Fig.S12, Fig.S13). We shall soon make our scripts available in GitHub.

8. About the estimator K-nearest neighbors (called here as Kraskov), how was the approach employed to find the best k by the authors? Did they try all possible values and choose the best one? How did they define or evaluate the best k?

Thank you for raising this important concern. Indeed the best k-parameter selection is very important for the Kraskov as the kernel width for the Kernel estimator. In our manuscript introduction Section we have mentioned the issues about the challenges involved in the estimation of information theory metrics with Kraskov, Kernel and Binning estimators for continuous data. We referred our earlier publication in Entropy which is also featured as a cover story about the quantification of information exchange in idealized and climate applications (https://www.mdpi.com/1099-4300/21/11/1094).

We proposed rigorous testing of the k-nearest and kernel width for consistency of the results in our previous publication. We followed similar principle in the current manuscript, for e.g., in the results section of idealized linear system, we mentioned that our free parameters i.e., kernel width and k-nearest neighbors are tested and tuned for consistent results with the test ranging from (20-60 neighbors) as well as (0.5-2 kernel widths). Similarly all our results are tested and tuned for consistent results. In addition, as the climate system data used in this study is near Gaussian, we also used robust linear estimator along sides with non-linear Kraskov and Kernel estimators.

In terms of code availability, it is a big plus and highly recommended that the authors publicly available their code in open source platforms (like GitHub, for instance). Therefore, other scholars and the community can use it to replicate the conclusions and compare it with their methods in future works.

Thank you, we agree. We will upload the codes in GitHub for public.

We agree with your concern regarding the physical atmospheric process dynamics.

We plotted the moisture transport anomalies (Fig. 6 in the manuscript) during the ENSO and IOD phases over the Indian domain. We 
[revised manuscript text omitted]

---

## Author Comment (AC3) · 7 Sep 2020

**Comment from Paul Pukite**

Dear Dr Paul Pukite,

Thank you very much for your comment. Here, you can find our answer.

Reviewer comments in: Black

Our reply in:  Blue

"Since it has become obvious that common-mode tidal forcing's control the majority of climate indices, as a first step one should consider how the tidal factors play into the models. See attached figures for AMO and ENSO. Once this causality is understood, then it will be much easier to deal with other interactions. ENSO and IOD have just slight variations on the tidal forcing.„

We agree with your argument on the common-mode forcing influence on the information sources and causal interactions. This is an issue in causality detection and information transfer quantification, where the common-mode forcing's on sources are usually ignored or misinterpreted. Our manuscript, entitled "Quantification of information exchange in idealized and climate applications" (https://www.mdpi.com/1099-4300/21/11/1094) precisely raised this concern on the attribution of causality. The manuscript concludes that care has to be taken while interpreting the results of information exchange/causality/correlation especially in case of common-mode forcing's.

But there is an ongoing discussion/debate on the causality between IOD and ENSO. However, the current manuscript deals with the interactions of IOD and ENSO on the Indian Monsoon Summer Rainfall (IMSR). With the methodology in the current manuscript, we could quantify the redundant of synergistic information exchange from sources i.e., IOD, ENSO to target, IMSR (please refer to the Methodology Section).

[Figure]

In the above figure (Figure 1 in the manuscript),

U represents unique information exchange from IOD and ENSO respectively to ISMR

R represents  redundant information from IOD and ENSO respectively to ISMR.

S represents synergistic information from IOD and ENSO respectively to ISMR.

Our method disentangles the interactions which are redundant or synergistic. If the IOD and ENSO share a common-mode forcing, then their influence on ISMR would be redundant. However, in our manuscript with the observational data, we see synergistic information exchange from IOD and ENSO to ISMR over Indian Monsoon core region (Figure 6 in our manuscript), which implies they are influencing ISMR tandemly (this shows they act as independent modes as far as their influence on ISMR is concerned). Establishing the causality between IOD and ENSO within the models needs a further detailed investigation which is beyond the research focus of the present manuscript.

Thank you again,
Authors.

---

## Author Response (AR1)

Dear Editor,

We would like to thank you for editing our manuscript and also our reviewers. Your comments helped us to improve our manuscript substantially.

We have included all the figures as mentioned in our response. In particular to your suggestion, we also believe the new figures we produced in our response to reviewer shall add value to our manuscript and hence we have included all the figures in supplementary material. In total now our supplementary material consists of 24 figures.

Here we are attaching the supplementary material and also the manuscript changes we made. The changes are highlighted in blue and red color.

In particular, we are also discussing point by point responses to the changes we made in supplementary material for your quick reference.

Supplementary material change:

1. Fig.1: Figure unchanged. Only minor changes with description, linear→Linear, kernel→ Kernel, kraskov→Kraskov to be consistent throughout the manuscript. The changes are highlighted and can be viewed in the attached manuscript below .

2. Fig.2: Figure unchanged. Only minor changes with description, linear→Linear, kernel→ Kernel, kraskov→Kraskov to be consistent throughout the manuscript. The changes are highlighted and can be viewed in the attached manuscript below .

3. Fig3: Figure unchanged, only added JJAS months in the description.

4. Fig4: Figure unchanged, only minor changes, I(PREC;ENSO), I(PREC;IOD),I(PREC;IOD,ENSO) changed to  *I(PREC;ENSO), I(PREC;IOD),I(PREC;IOD,ENSO)*

5. Fig5: Figure added as per the suggestion of Reviewer

6. Fig6: Figure unchanged. Only minor changes with description, linear→Linear, kernel→ Kernel, kraskov→Kraskov to be consistent throughout the manuscript. The changes are highlighted and can be viewed in the attached manuscript below .

7. Fig7: Figure shifted due to an extra added figure i.e., Fig5

8. Fig8-Fig13,  New figures added as per the editor suggestion to include the figures from Author response to reviewer.

9. Fig14-Fig16: Figures unchanged but shifted due to additions of new figures Fig8-Fig13.

10. Fig 17-Fig19: New figure added as per the suggestion of Reviewer.

11. Fig20-Fig.24: Figures unchanged but shifted due to additions of new figures

Manuscript Changes:

We added our GitHub path for the users to access our codes as per reviewer suggestion.  We also acknowledged our reviewers in acknowledgment section.

[Figure]

**Fig. 1** Information exchange in nats from two-source (red line), single source (green and blue lines), and net synergy (black line) to target for Linear,  Kraskov and  Kernel estimators. The error bars represents two standard deviations of the 100 permuted samples.

[Figure]

**Fig. 2** Information exchange in nats from two-source (red line), single source (green and blue lines), net synergy (black line) to target for Linear,  Kraskov and  Kernel estimators. The error bars represents two standard deviations of the 100 permuted samples.

[Figure]

**Fig. 3** Regressions of PCs obtained from their respective EOFs over the Indian and Pacific Oceans with the observed IOD and Niño 3.4 Index and their associated percentage contribution to the total variance for HadISST and NCEP reanalysis SST data sets for JJAS.

[Figure]

**Fig. 4** Information exchange from  $I(PREC; IOD)$,  $I(PREC; ENSO)$, two-source information exchange  $I(PREC; ENSO, IOD)$ and NET SYNERGY $\times 10^{-2}$ nats for observational data sets GPCC, APHRODITE and NCEP reanalysis with Kraskov estimator for JJAS. Only significant values at 95% confidence intervals are plotted.

[Figure]

**Fig.** **5** Information exchange from $I(PREC; IOD)$, $I(PREC; ENSO)$, two-source information exchange  $I(PREC; ENSO, IOD)$ and NET SYNERGY $\times 10^{-2}$ nats for observational data  sets GPCC, APHRODITE and NCEP reanalysis with Kernel estimator for JJAS. Only significant values at 95% confidence intervals are plotted.

[Figure]

**Fig.** **6** Information exchange from $I(PREC; IOD)$, $I(PREC; ENSO)$, two-source information exchange  $I(PREC; ENSO, IOD)$ and NET SYNERGY $\times 10^{-2}$ nats for observational data set ERA Interim reanalysis (1980-2005) for JJAS. Only significant values at 95% confidence intervals are plotted.

[Figure]

**Fig. 7**  Information exchange from $I(PREC; IOD)$,  $I(PREC; ENSO)$, two-source information exchange $I(PREC; ENSO, IOD)$ and  NET SYNERGY $\times 10^{-2}$ nats for  observational data set MERRA-2 reanalysis ( 1980-2005) for JJAS. Only significant values at 95% confidence intervals are plotted.

[Figure]

**Fig. 8**  EOF2 patterns of  SST anomalies (DJFM) in the Indian ocean and EOF1 patterns in the Pacific ocean for observed HadISST and NCEP reanalysis.

[Figure]

**Fig. 9** SST composites  (DJFM) in the Indian ocean and  the Pacific ocean for observed HadISST.

[Figure]

**Fig. 10** SST composites (DJFM) in the Indian ocean and the Pacific ocean for observed NCEP reanalysis.

[Figure]

**Fig. 11** Regressions of PCs obtained from their respective EOFs over the Indian and Pacific Oceans with the observed IOD and Niño 3.4 Index and their associated percentage contribution to the total variance for HadISST and NCEP reanalysis SST data sets for DJFM.

[Figure]

**Fig. 12** Total precipitation anomaly (mm/month) composites (DJFM) over the Indian subcontinent for El-Niño, La-Niña, positive IOD and negative IOD events observed in GPCC, APHRODITE and NCEP reanalysis data sets for the period of 1951-2005

[Figure]

**Fig. 13** Information exchange from $I(PREC; IOD)$, $I(PREC; ENSO)$, two-source information exchange $I(PREC; ENSO, IOD)$ and NET SYNERGY $\times 10^{-2}$ nats for observational data sets GPCC, APHRODITE and NCEP reanalysis for DJFM with Linear estimator. Only significant values at 95% confidence intervals are plotted.

[Figure]

**Fig. 14** Percentage of the total variance contributed by the first 20 EOFs to the total variability in Indian and Pacific Ocean SST for MPI-ESM-LR, Nor-ESM-M and EC-EARTH models for the month of JJAS (1951-2005)

[Figure]

**Fig. 15** SST composites for observations and GCMs for various phases of IOD events over the Indian ocean for JJAS.

[Figure]

**Fig. 16** SST composites for observations and GCMs for various phases of ENSO events over the Pacific ocean for JJAS.

[revised manuscript text omitted]

215 **2.3.2**

We also extended our analysis to a non-linear  Heńon

220  system described in the Appendix section.

**3  Data and climate models**

In this section, we will discuss various observational and reanalysis data sets used to quantify the two-source IE from ENSO and IOD to IMSR interannual variability in the natural system. Furthermore, the details of various GCM and RCM simulations used in this study are also covered.

**3.1  Observational, reanalysis data sets and climate simulations**

We are focusing on the South Asian Summer Monsoon seasons, starting from June and ending in September (June- July-August-September: JJAS), thus monthly data sets for JJAS for the time period 1951-2005 from observations and model simulations are used in this study. Various observational, reanalysis data sets and model simulations used to quantify the two-source IE from the ENSO and IOD to the ISMR interannual variability are listed in Table 1 and are also described here.

**3.1.1  Observational, reanalysis data sets and indices**

The UK Met Office's Hadley Centre Sea Ice and Sea Surface Temperature dataset (HadISST 1.1) (Rayner et al., 2002) is used to retrieve SST information for the Indian and the Pacific ocean. Monthly precipitation fields from Global Precipitation Climatology Centre (GPCC) (Schneider et al., 2008) is used as precipitation observational record together with a high-resolution data set, covering only the monsoon south Asia domain, namely the Asian Precipitation - Highly-Resolved Observational Data Integration Towards Evaluation (APHRODITE) monthly accumulated precipitation (Akiyo et al., 2012). The rainfall, winds, and specific humidity are taken from the National Center for Environmental Prediction–National Center for Atmospheric Research (NCEP–NCAR) reanalysis data set (Kalnay et al., 1996). The ENSO and IOD indices are obtained from the National Oceanic and Atmospheric Administration Earth System Research Laboratories(NOAA ESRL) and Japan Agency for Marine-Earth Science and Technology(JAMSTEC) for validation of PCs derived from the observational SST data sets, i.e., the HadISST, and NCEP reanalysis SST. In addition to the above-mentioned data sets, we also used ERA-Interim (Dee et al., 2011) and MERRA (Rienecker et al., 2011) reanalysis rainfall datasets (1980-2005) as additional resources.

**3.1.2  Global and regional climate simulations**

The three CMIP5 GCMs (details in Table. 1), the MPI-ESM-LR (Stevens et al., 2017), Nor-ESM-M (Bentsen et al., 2012) and EC-EARTH (Hazeleger et al., 2010) were dynamical downscaled with the non-hydrostatic regional climate model COSMO-crCLM version v1-1. The COSMO-crCLIM is an accelerated version of the COSMO model (Fuhrer et al., 2014) in climate mode (Leutwyler et al., 2016; Rockel et al., 2008). A two-stream radiative transfer calculations are based on Ritter and Geleyn (1992), the convection is parameterized by Tiedtke (1989), the turbulent surface energy transfer and planetary boundary layer are using the parametrization of Raschendorfer (2001), and precipitation is based on a four-category microphysics scheme that includes cloud, rainwater, snow, and ice (Doms et al., 2011). The soil-vegetation-atmosphere-transfer is using the TERRA-ML (Schrodin and Heise, 2002), however, this current version is employing a modified groundwater formulation (Schlemmer et al., 2018). The RCM simulation has a horizontal resolution of $0.22°$ (i.e., 25km) and with 57 vertical levels and is using

**Table 1.** CMIP5–GCMs/RCM/observations descriptions used in the current study.

[revised manuscript text omitted]